# Modeling Student Learning with 3.8 Million Program Traces

## Abstract

As programmers write code, they often edit and retry multiple times, creating rich "interaction traces" that reveal how they approach coding tasks and provide clues about their level of skill development. For novice programmers in particular, these traces reflect the diverse reasoning processes they employ to code, such as exploratory behavior to understand how a programming concept works, re-strategizing in response to bugs, and personalizing stylistic choices. In this work, we explore what can be learned from training language models on such reasoning traces: not just about code, but about coders, and particularly students learning to program. We introduce a dataset of over 3.8 million programming reasoning traces from users of PENCIL CODE, a free online educational platform used by students to learn simple programming concepts. Compared to models trained only on final programs or synthetically-generated traces, we find that models trained on real traces are stronger at modeling diverse student behavior. Through both behavioral and probing analyses, we also find that many properties of code traces, such as goal backtracking or number of comments, can be predicted from learned representations of the students who write them. Building on this result, we show that we can help students recover from mistakes by steering code generation models to identify a sequence of edits that will results in more correct code while remaining close to the original student's style. Together, our results suggest that many properties of code are properties of individual students and that training on edit traces can lead to models that are more steerable, more predictive of student behavior while programming, and better at generating programs in their final states[1].

## 1 Introduction

Imagine a student who learns to code with a simple visual programming assignment, such as drawing a snowman (Figure 1). At first, they might try to figure out how to draw a single circle, and explore setting different parameters, such as its radius. Next, the student may make a straightforward attempt towards completing the rest of the snowman's body. After executing the program (and seeing an unexpected result), the student might realize that they do not fully understand how 2D coordinates work, and ask for help. Once the hardest parts of the program have been worked out, the student may finish by personalizing the program, adding comments, or modifying colors until they feel satisfied. Any system (or human reader) who sees only this final program state can infer only a limited part of the reasoning process the student engaged in while programming.

Despite the diverse set of strategies and behaviors users engage in when problem-solving, very few public sources of data provide glimpses into this underlying process.[2] This poses a fundamental gap in the standard paradigm of leveraging internet-scale data for pretraining large language models: current data may be used to successfully model what content users produce, but not *how* they created it. Therefore, state-of-the-art models have been shown to often take non-human like approaches to problem solving (McCoy et al., 2024; He-Yueya et al., 2024). And while there has been a recent surge in interest in training large language models to produce "reasoning" traces (Huang & Chang, 2023; dee, 2025), these methods have primarily been developed for the goal of improving model performance, rather than accurately modeling human reasoning.

---

[1]Code and data is available at [REDACTED]

[2]Examples include StackOverflow, Wikipedia Revision History, and r/learnprogramming.

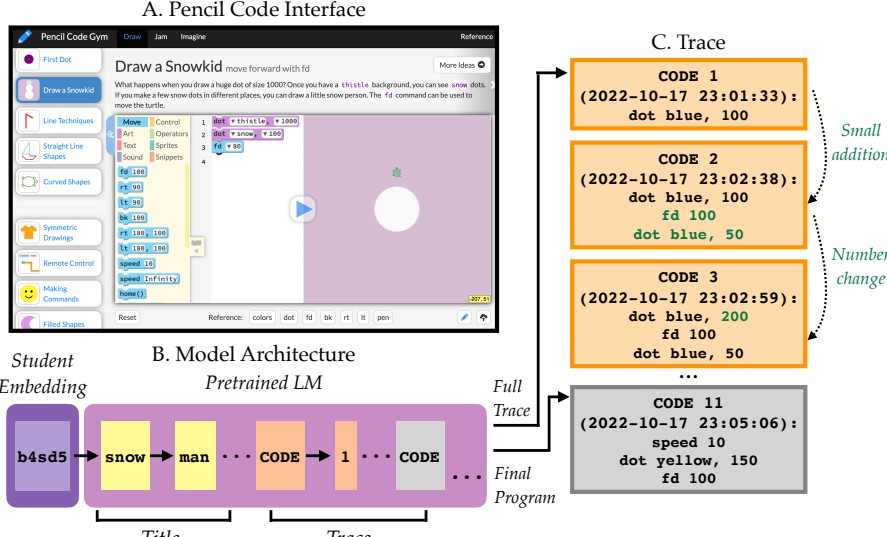

Figure 1: (A) User interface of PENCIL CODE, where users can program with visual block coding. (B) Overview of the model architecture we train, with an embedding layer for student IDs. (C) An example trace written for title snowman, along with edit types.

However, a major paradigm shift is emerging in how users are interacting with AI assistants, especially code generation models. Tools such as Cursor, OpenHands, and GitHub CoPilot can seamlessly integrate into programmers' development environments, thereby capturing rich interaction data that records a user's entire coding process, including their attempts, revisions, and interactions with code assistants. What new insights can code generation models learn from such data? Can we learn more about the difficulties faced during the development process, and even model the different ways individuals approach the same programming task? Understanding an individual's entire reasoning process when programming is particularly useful in education contexts, where student iterative debugging and refactoring behaviors play an important role in learning (Berland et al., 2013).

In this paper, we first curate a dataset of 3.8 million *program traces* from real students learning to code on PENCIL CODE, an educational programming platform. Our dataset spans 9 years of usage from more than 1 million unique students solving a rich variety of assignments, ranging from standard graphical assignments like snowman to more complex functions implementing search algorithms. Each program trace consists of a unique student ID and title, as well as a temporally ordered sequence of program states the student executed. While prior works created datasets capturing IDE-level edit actions (Brown et al., 2018) or final code submissions (Eliseeva & Koutcheme, 2023), to the best of our knowledge, ours is the first large-scale dataset of execution-bounded code edit sequences across diverse assignments suitable for language model training.

We then compare three different language model pre-training approaches: training on program traces directly (trace), training on traces that are synthetically generated based on the final program state (synthetic), and training only on the final program state (last). We use both ***behavioral*** (studying generated samples) and ***representational*** (probing learned embeddings) evaluations to study how well these different LMs learn to model both final program states and the process that produced them.

Our findings demonstrate that training on traces leads to a richer model of both programs (§4.1) and student behaviors (§4.3), compared to models not trained on ground truth traces. We also find that representations of student IDs encode nontrivial information about many properties of their program traces, including those reflective of their reasoning process (*e.g.,* the frequency of deviation from the goal state, time spent on a particular assignment), as well as stylistic aspects of their code (*e.g.,* number of comments; §4.2, §4.3). Furthermore, many of these properties can be learned about a student efficiently, with only a few modified parameters and trace examples from a student (§4.4). Lastly, we show how to leverage learned properties to help students recover from errors (§4.5).

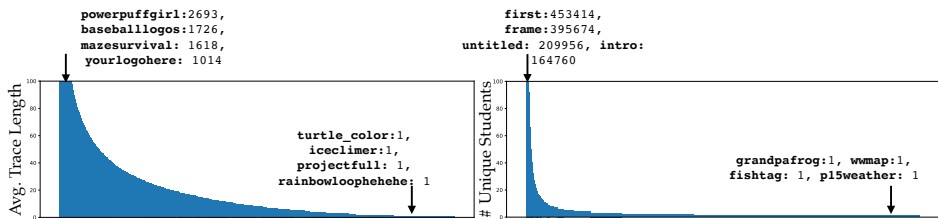

Figure 2: Distribution of average student program trace length (left) and number of unique students (right) for programs in the PENCIL CODE dataset, showing a heavy long tail.

## 2 PENCILCODE OVERVIEW

PENCIL CODE,[3] introduced by Bau et al. (2015), is an open-source educational platform for programming that supports creative projects spanning turtle graphics, music composition, speech synthesis, networking, and interactive storytelling. It utilizes Droplet, a dual-modality code editor that allows users to write code through either a visual block-based interface (similar to Scratch) or directly in web programming languages like CoffeeScript, JavaScript, HTML, and CSS. The platform's block-and-text interface has been shown to support novice programmers' development of expertise (Blanchard et al., 2020) and to enhance students' programming skills and attitudes (Blanchard et al., 2019; Deng et al., 2020). Unlike past work that studies student programming behaviors with a limited, structured set of assignments (Piech et al., 2015b), PENCIL CODE facilitates studying a more diverse range of student behaviors, including intrinsically-motivated exploration and practice.

**Statistics** We construct a dataset of 3.8M programming traces written on PENCIL CODE from 2015 to 2024. Each trace consists of a hashed student ID, title (*e.g.,* snowman), and an ordered of sequence of programs written by the student, along with associated timestamps. The overall dataset has size 248GB, consisting of 1.3M unique usernames and 3.8M unique (username, program_name) pairs. There is an average of 2.86 program traces per user, and the trace titles with the highest and lowest frequencies and average trace length are shown in Figure 2.

## 3 EXPERIMENTAL SET-UP

### 3.1 MODELS

We train 5 models on PENCIL CODE data by continued pretraining of LMs. Experiments reported in the main paper are conducted with a base 124M parameter `GPT-2` model (Radford et al., 2019). However, we also run experiments with a 1B parameter `OLMo-2` model (OLMo et al., 2024), for which we observe similar results, reported in Table 3. The **trace** model is trained on full traces, while the **last** model is trained on only the last program in each trace. Building on prior work (Piterbarg et al., 2025), we also train a **synthetic** model that is trained on traces that are synthetically generated based on the last program in each trace.[4] In addition, we train **trace downsampled** and **synthetic downsampled** models on versions of the trace and synthetic datasets that are downsampled to match the same number of unique tokens as in the last dataset (details in §A.3).

Building on prior work steering language models with personalized embeddings (Zhong et al., 2021; Doddapaneni et al., 2024), we train models with a **student embedding** layer that maps a student ID to a 768-dimension embedding, which is introduced as a "soft token" at the start of all program sequences (Figure 1). Introducing an explicit student token enables analysis of behavior at the student level (§4.3), as well as light-weight adaptation to new students (§4.4).

---

[3]pencilcode.net

[4]Piterbarg et al. (2025) construct synthetic traces by assuming each edit adds a set of sequentially dependent lines, which may not actually reflect the types of edits students make while learning how to program. We similarly construct synthetic traces by assuming each edit adds a new PENCIL CODE instruction (e.g., `fd 20`), while ensuring that no synthetic trace is longer than the original program trace.

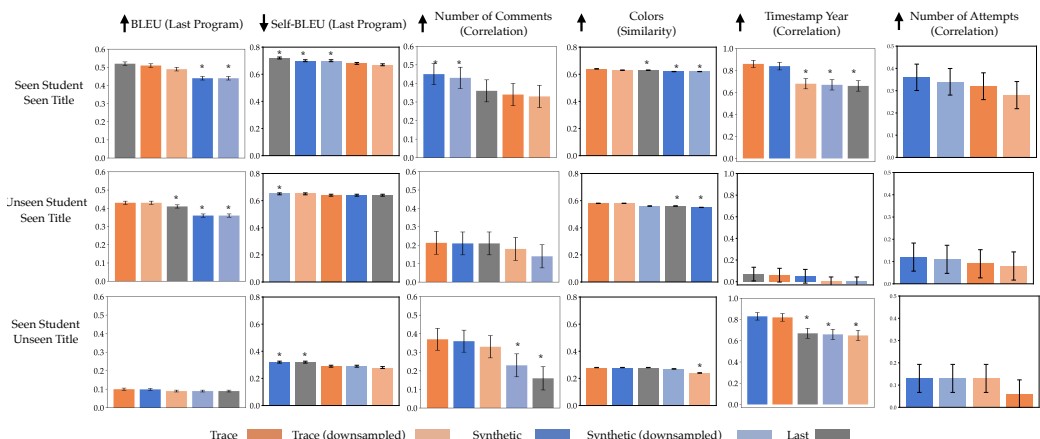

Figure 3: **Correlation of Generated Trace Properties with Ground Truth (Final Program State)**
We evaluate the generated final program state of a trace from sampling all models across evaluation
splits. Correlation denotes Pearson's correlation coefficient. The Colors metric compares the cosine
similarity of between program color embeddings. * indicates a statistically significant difference
with the `trace` model using a paired T-test between unique (student, title) pairs at $p = 0.05$ with
Bonferroni correction, and error bars indicate standard errors of the mean.

## 3.2 DATASET FORMATS

**Evaluation Splits**   We study various kinds of generalization of the above models (§4.1): general-
ization in-distribution to new (student, title) pairs (where each has been seen before separately), as
well as out-of-distribution generalization to unseen students and titles. We create 4 test sets reflecting
each kind of generalization. We first hold out 2% of student IDs and 2% of trace titles. We then
take 80% of the non-held-out data and designate them as training data, then create the in-distribution
test set by taking all remaining non-held-out data. We refer to this split as `seen student/seen`
`title`. We then create 3 additional test sets that differ in whether the student or title is held out: `seen`
`student/unseen title`, `unseen student/seen title`, and `unseen student/unseen title`.
See §A for more details, including the sizes of each split. Additionally, many of the trace titles in
the PENCIL CODE dataset are based on real-world assignments from learning materials. For some
analyses, we look specifically at a subset of trace titles that correspond to such assignments.

## 3.3 EVALUATION METHODS

We consider two types of evaluations of our 5 models: The first is a **behavioral** evaluation, where we
generate Monte Carlo samples with a model and analyze properties of the generated programs (§4.1,
§4.4, §4.5).[5] The second evaluation type is **representational**, where we probe learned code and
student embeddings to understand what information they encode (§4.2, §4.3). For both, we analyze a
variety of properties of code written by students. For more details, see Appendix D.

**Properties of Programs**   For each program that is part of a trace, we measure **successful execution**
(whether a program executes without errors), the **time** the program was executed on the PENCIL
CODE server, and the number of occurrences of certain **keywords** (*e.g.,* the word `turtle`, which
is associated with *turtle graphics* programs, words associated with *colors* such as `magenta`), and
*comments* (*e.g.,* lines starting with #). We measure these properties for each program in a trace,
although we are particularly interested in the last program (which we take as the student's goal state).

**Properties of Traces**   We also analyze properties of traces, which characterize the full sequence of
program code and edits. For the program metrics above, we can consider the mean value across all
programs in a trace. In addition, we can look at trace-specific properties including **goal backtracking**:

---

[5]For all behavioral evaluations, we generate program traces (or single programs from the `last` model) using
nucleus sampling with $p = 0.9$ (Holtzman et al., 2020). For details on how we parse generations, see App. E.

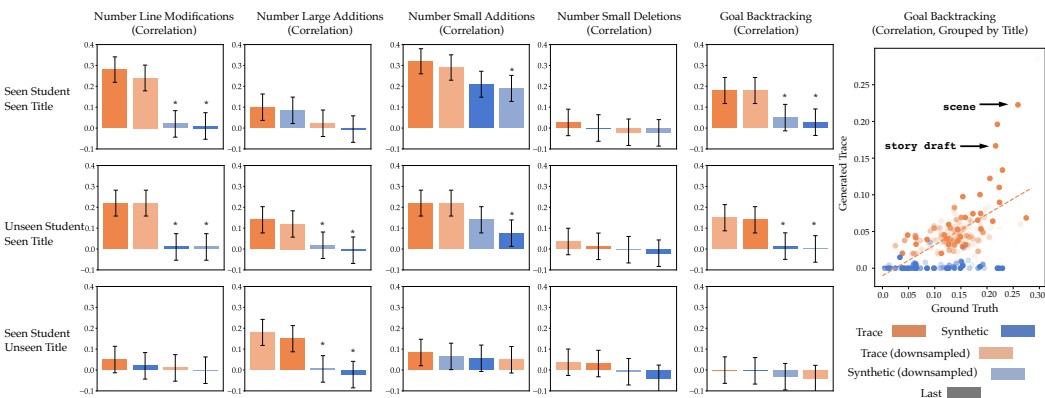

Figure 4: **Correlation of Generated Trace Properties with Ground Truth (Full Program Trace)** We evaluate full generated program traces by sampling all models except `last` across evaluation splits. Correlation denotes Pearson's correlation coefficient. * indicates a statistically significant difference with the `trace` model using a paired T-test between unique (student, title) pairs at $p = 0.05$ with Bonferroni correction. Error bars indicate standard errors of the mean. The right scatter plot highlights how the `trace` model successfully captures student goal backtracking, with higher correlation for trace titles that are common in the training data (higher opacity).

We measure the *goal backtracking ratio* of a trace, which is the average fraction of times in a trace that a student's edit results in an *increase* in edit distance between the current program and goal program state. We also measure the counts of **edit types** in a trace: small/large additions, small/large deletions, color/number changes, and comment/function additions.

**Behavioral Evaluation Metrics** While for our representational evaluations (§4.2, §4.3) we can directly measure a probe model's ability to predict the properties of code listed above, for our behavioral evaluations, we *compare* generated samples against ground truth traces written by students. We report the Pearson **correlations** between these values for each metric. We additionally measure the **BLEU** score (Papineni et al., 2002) to directly compare the similarity of generated program traces against the ground truth trace for a given student ID and title, as well as the **Self-BLEU** (Zhu et al., 2018) across the final programs of repeated generated samples. Whereas BLEU captures how close a program is to a reference, Self-BLEU measures how similar a set of generated samples is, with a lower value indicating higher diversity. For both metrics, we average across $1, 2, 3, 4-$ngram scores.

## 4 EXPERIMENTS

First, we investigate whether models trained on real edit traces learn richer representations than those trained on synthetic traces or final programs (Sections 4.1-4.3). Second, we examine whether these representations encode student-specific information that enables efficient personalization (Section 4.4). Finally, we demonstrate how learned representations can be applied to practical educational scenarios like style-preserving error recovery (Section 4.5).

### 4.1 MODELING GENERAL BEHAVIOR

We first ask whether code generated by models is reflective of the programming behaviors of PENCIL CODE students. We select 200 titles that correspond to assignments found in online resources from PENCIL CODE.[6] For each title, we randomly select 50 students, split evenly between the seen and unseen student splits. We then use the student ID and title to construct a prefix which we use to conditionally generate $n = 20$ random samples from each model. We then analyze each sampled trace (or single program in the case of the `last` model) for properties described in Section 3.3.

---

[6]We derive them from external resources on learning to code in PENCIL CODE, including an associated primer (https://book.pencilcode.net/) and teaching materials (https://guide.pencilcode.net/), which contain several standard introductory assignments. 100 of these were seen during training, and 100 were not.

Computing metrics on final program states, such as those in Figure 3, requires being able to parse a final program state from a generated trace. However, the trace models occasionally do not generate the end of sequence token; in such cases, we take the last full generated program state (separated by `CODE n`).[7] This approach creates a harder evaluation setting for the trace models than for the `last` model because the parsed states from the trace models' generations may not be final; therefore, for an additional comparison, we present results that filter out generations that do not reach the end of sequence token in Appendix F.

Lastly, while we present results for a `GPT-2` model in this section, Table 3 in Appendix C shows that we see similar generalization trends for a 1B `OLMo-2` model. Across all splits, the `trace downsampled` model generates final program states that are more similar to ground truth generated programs than those generated by the `last` model, suggesting that the gap between trace and `last` model increases with scale.

**Generalization In-Distribution**    As shown in Figure 3 (first row), for students and titles that were seen separately during training (but not together), the `trace` model generates final programs that are more similar (BLEU[8]) to the ground truth than the `synthetic` model and comparable to the `last` model. However, the `trace` model results in higher diversity between generated last programs (lower self-BLEU score) than both the `last` and `synthetic` models. These results hold even when we control for the number of tokens (`trace downsampled` and `synthetic downsampled`), suggesting that training on traces can leader to stronger and richer final program states.

**Generalization Out-of-Distribution**    The second and third rows in Figure 3 present results analyzing the generated final programs for splits where either the student or trace title was unseen. These results reveal how models leverage student IDs and titles: for example, knowing the student ID leads models to generate programs with the correct timestamp years, while knowing the title (*e.g.,* rose) is sufficient for using the appropriate colors in graphical programs. We observe that BLEU significantly drops across all models for the `seen student/unseen title` split; thus, despite models' natural language pretraining (prior to training on PENCIL CODE), generalization is still difficult.[9]

**Student Edit Behavior**    Finally, we ask if generated program traces reflect students' *edit* behavior. Figure 4 shows that goal backtracking in generated traces from the `trace` model are indeed correlated with ground truth metrics, but this correlation slightly decreases when the student is unseen. This suggests that title and student ID *both* play a role in determining whether there is backtracking from the final program in the trace. We also see in the rightmost scatter plot that the correlation of degree of backtracking between generated and ground truth traces is higher for titles that are more frequent in the training data (*e.g.,* scene). As expected, the `synthetic` model only shows high correlation for the "small addition" types of edits, which are the only kind it sees during training.

### 4.2    PROBING CODE REPRESENTATIONS

§4.1 presents evidence that the `trace` model can predict aspects of students' behavior in its generations. In this section, we ask whether such information is directly encoded in the model's representations of code: Given a code snippet written by a student, can a probe trained on the `trace` model's representations predict *future* student behavior, *e.g.,* whether the student will backtrack from their goal in the future? To what extent is the performance of the probe (and therefore the information in code representations) dependent on the model's learned information about the particular student?

We construct our probing dataset by first considering the same restricted set of 100 PENCIL CODE assignment titles that were seen during training (as described in §4.1). For each title, we randomly sample 50 traces from the `seen student/seen title` split.[10] Additionally, we only analyze students

---

[7]See §E for more details on how we sample generations.

[8]We use BLEU rather than CodeBLEU (Ren et al., 2020) as we found it to be more discriminative between pairs of different Pencil Code programs in pilot experiments.

[9]We hypothesize that this is likely due to many trace titles not reflecting program semantics (*e.g.,* somethingforclass), as we observed qualitatively that titles in this split that *do* reflect program semantics (*e.g.,* spike function result in uniformly high BLEU scores, suggesting some degree of generalization.

[10]We filter out any program names that do not have at least 50 traces in the split.

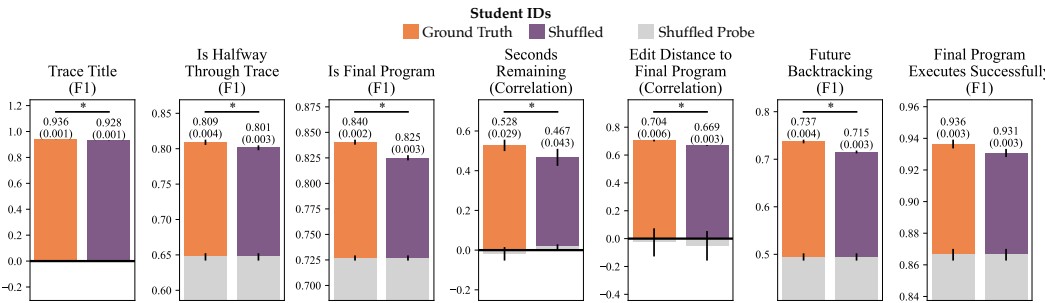

Figure 5: **Probing Code Representations Given Student IDs and Trace Prefixes.** We report mean F1 scores / Pearson correlations for probes trained on 5 random data splits. Error bars indicate standard errors of the mean. *Shuffled Probe* corresponds to the control where we shuffle inputs/outputs to the probe. * indicates statistically significant differences between the ground truth student and shuffled student probes under a paired T-test between random probes. See §4.2 for details.

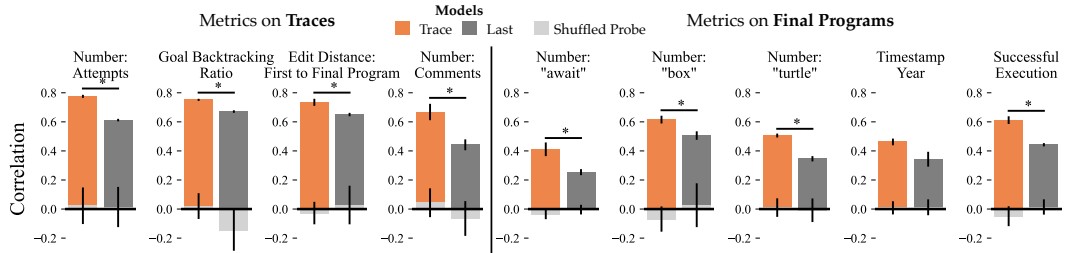

Figure 6: **Probing Student Representations to Predict Means Across Traces for a Student.** We train probes to predict, for a given metric and student, the mean value of the metric across all of the student's traces. We report mean Pearson correlations for probes trained on 5 random data splits. Error bars indicate standard errors of the mean. *Shuffled Probe* corresponds to the control where we shuffle inputs/outputs to the probe. Results are shown for metrics on final programs and trace metrics. * indicates statistically significant differences between the ground truth student and shuffled student probes under a paired T-test between random probes. See §4.3 for details.

with between 20 and 200 traces in the training dataset for the `trace` model to ensure that sufficient traces were seen to learn meaningful representations.[11]

We then construct a probing dataset where inputs are embeddings of traces consisting of varying numbers of programs, and outputs are various code properties. For details, see §I. We train ridge regression/classification probes to predict the *trace title*,[12] whether the program *is the last program*, and whether the program is at least *halfway through the trace*. We also predict more fine-grained future behavior from the student, including whether the student will *backtrack* from the goal/final program later in the trace, the number of *future attempts* the student will make, the number of *seconds* the student will continue to spend, the *edit distance* between the current program state and the final program state, and the eventual correctness of the final program in the trace. See §I for more details.

To evaluate the impact of student IDs, we construct input embeddings from shuffled student IDs. We also shuffle the embeddings themselves as a control task for how well probes perform when not using the actual representations (Hewitt & Liang, 2019).

**Results** Figure 5 shows mean F1 scores / Pearson correlations for probes trained across 5 random train/test splits. We find that for all metrics, probes significantly outperform the shuffled probe controls, suggesting that the embeddings contain nontrivial information about code and future student behavior. We also find that for most metrics, conditioning on the ground truth student ID leads

---

[11]Students with more than 200 traces appeared to be classrooms with multiple students using the same account.

[12]For this metric, we mask program titles in constructing inputs, *i.e.,* we set $a$ to the mask token

to statistically significant better performance than conditioning on shuffled student IDs (except for the successful execution of the final program). Our results suggest not only that the `trace` model leverages information about individual students in reasoning about future code in a trace, but also that its embeddings could be used to enable supportive educational feedback, *e.g.,* intervening when a student is about to backtrack from their goal.

### 4.3 PROBING STUDENT REPRESENTATIONS

§4.1 and §4.2 show both behaviorally and representationally that the `trace` model uses information about student IDs to make predictions about their behavior. In this section, we directly probe student representations to evaluate what they capture about a student's programming style and abilities.

We construct our student probing dataset as follows. Let $S$ denote the set of students, where we randomly sample $|S| = 2000$ students, each with between 20 and 200 traces in the training datasets for the `trace` and `last` models. For each student $s \in S$ and property of interest $y$, we compute a ground truth "student-specific" target value $y^s$ by averaging the value of the metric across all traces written by $s$. We construct a student embedding $e^s$ for each $s$ by extracting the learned embedding vector for $s$ from the `trace` or `last` model. We construct the probing dataset $\mathcal{D}_{\text{student}} = \{(e^s, y^s) : s \in S\}$. We train MLP probes on 5 random train/test splits of $D$. As in §4.3, we train probes on datasets where the student embeddings $e^s$ are randomly shuffled with respect to $y^s$ to serve as a control for how well probes perform when relying purely on statistics in the target metric (Hewitt & Liang, 2019).

**Results** As shown in Figure 6, for both the `trace` and `last` model, probes trained on ground truth student representations outperform the probes trained on shuffled student representations, suggesting that the student representations in both models encode nontrivial information about students. We find, however, that the `trace` model's student representations consistently lead to statistically significant better performance than the `last` model's student representations (except for timestamp year), highlighting that the `trace` model has learned about student behavior beyond what can be learned just from the kinds of programs students write. Even for metrics directly related to the kinds of programs students write (*e.g.,* occurrences of `await`), which could be learned by the `last` model, the `trace` model's student representations lead to better performance, suggesting that by training on traces, the `trace` model has learned richer information about students.

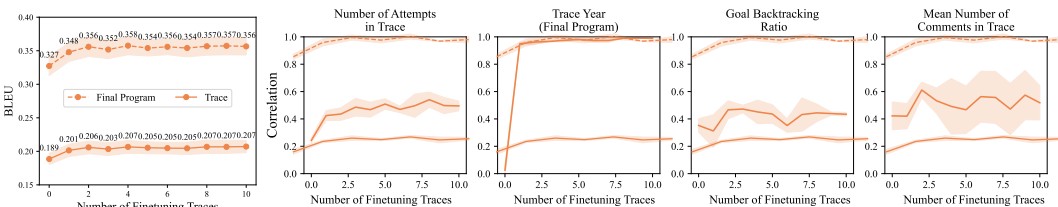

Figure 7: **Adaptation to New Students.** The leftmost plot shows BLEU score, and the right plots show correlations for different code properties. Results are for finetuning just student embeddings of the `trace` model. We report means and standard errors of the mean across 3 random seeds.

### 4.4 ADAPTING TO NEW STUDENTS

Although our model cannot predict behavior for completely new students from IDs alone, handling new students efficiently through few-shot personalization is crucial for practical deployment in educational settings where new learners continuously join the platform. Recall that a new student ID is first mapped to a random 768-dimension via the student embedding layer. We evaluate how efficiently – in terms of both data and model parameters – the `trace` model can adapt to new students. We first select all unseen users from the `unseen student/seen title` split who have between 20 to 200 traces in the split. We then split these students into two groups: 5% for hyperparameter selection and 95% for finetuning and evaluation. For each student, we order their traces from earliest to latest, and for each $k \in [1, ...10]$, we train on the first k traces of each student, then evaluate on the

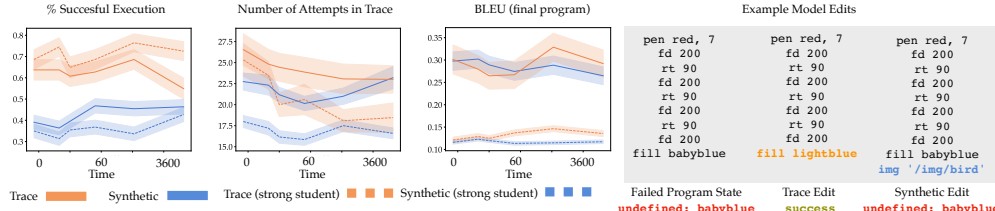

Figure 8: **Model Controllability and Error Recovery** We sample program traces from the `trace` and `synthetic` models by conditionally generating on prefixes consisting of randomly sampled program states that failed to execute, held-out from training. We vary the amount of time between the timestamp on the broken trace and the next program, as well as compare the effect of student embedding (dashed vs. solid). Shaded area represents standard error of mean, and we present example edits from both models for a random failed state ("babyblue" is an undefined color).

remaining traces.[13] We hold all parameters fixed except the student embedding layer, updating only the weights of the student MLP module associated with the students whose traces are finetuned on. See Appendix G for more details.

**Results** As shown in Figure 7, there is an increase in BLEU score for both final programs and full traces with just a single finetuning trace per student. We observe that BLEU scores stagnate after around k=4 finetuning traces. We also observe a sharp increase in correlation for trace year, followed by stagnation, suggesting that this property is learned quickly. For number of attempts in a trace, correlation increases, while improvements for mean number of comments and goal backtracking ratio are more mixed, with correlations increasing slightly until k=2, then stagnating.

## 4.5 ERROR RECOVERY AND MODEL CONTROL

Finally, we evaluate the ability of the `trace` model to help students recover from errors and investigate whether these abilities can be controlled. We sample mid-trace programs that fail to execute written by held-out students and create inputs with their student ID, the trace title, the sequence of program states up until that program, and the time header for the next program state (*e.g.,* CODE 6:(2018-10-19 14:12:38). We then conditionally generate the remainder of the trace using either the `trace` or `synthetic` model and measure: whether the final program state successfully executes, the number of attempts taken, and the BLEU score between the generated and ground truth final programs.

Additionally, we compare the effect of two kinds of model controllability on error recovery. First, we vary the time header in the prefix: If the failed program has the header CODE 5: 2018-10-19 14:12:37, we select a time $t \in [0.5, 1, 5, 60, 6400]$ in seconds which we add to the header (*e.g.,* CODE 6: 2018-10-19 14:12:38 for $t = 1$). Second, we vary replacing the ground truth student ID with a "strong student" embedding, selected by choosing the student in our training dataset with the highest number of program traces and lowest degree of goal backtracking. Additional details are in §J.

**Results** In Figure 8, we find that the `trace` model generates traces that reach a successful program state more than **60%** of the time, outperforming the `synthetic` model, which can only append lines as edits. Furthermore, replacing the student ID with the strong student embedding (dashed lines) increases the rate of successful execution of generated final programs, but only for the `trace` model, confirming that the student embedding carries useful information about strong student edit behavior.[14] Finally, although increasing the amount of time between the failed program state and the rest of the trace does not appear to affect successful code execution, it leads to fewer numbers of attempts for the `trace` model. This indicates that we can control the granularity and extent of program modifications, which can be useful for adapting to different styles of feedback and interventions.

---

[13]To ensure fair comparison across different values of k, we evaluate on the set of examples that have been held out for all k values.

[14]Interestingly, we observe the opposite when evaluating the BLEU score: using the strong student embedding decreases the ability for both `trace` and `synthetic` models to reach final program states that are similar to the ground truth, showing that the learned student embeddings carry important information for personalization.

## 5 RELATED WORK

Motivated by concerns that AI tools such as large language models (LLMs) might foster student over-reliance by reducing student engagement (Bastani et al., 2024; Nie et al., 2024), there has been increasing interest in understanding how to effectively use LLMs within educational contexts, such as generating debugging hints in programming contexts (Kotalwar et al., 2024; Phung et al., 2024). These works build on an extensive line of research that studies how computational modeling tools can successfully capture how student knowledge evolves over time (Corbett & Anderson, 1995; Piech et al., 2015a; 2012; Gao et al., 2025), as well as identify different stages of student behaviors when learning programming, such as "tinkering" versus planning (Berland et al., 2013).

Closest to this work are papers that propose methods to learn "edit embeddings" from student code edits; however, they largely consider smaller datasets that do not contain student behavior across assignments as diverse as in PENCIL CODE (Piech et al., 2015b; Heickal & Lan, 2025). Furthermore, these works do not address whether features useful for personalized edit modeling can only be learned from real student edit behavior, versus the `synthetic` or `last` baselines we compared against. Other datasets capturing student programming behavior include BlueJ Blackbox (Brown et al. (2018)), which captures extremely granular IDE-level actions; FalconCode (Eliseeva & Koutcheme (2023)), which contains only final submissions rather than intermediate states; and StudentEval (Khoja et al. (2024)), which captures student prompts rather than actual coding behavior. Finally, concurrent work by Miroyan et al. (2025) evaluates how LLMs finetuned on a much smaller dataset (< 1M traces) generates traces aligned with real student traces along several properties of code (*e.g.,* error patterns). However, they do not go beyond surface-level alignment or disentangle which code properties can be learned as features of students versus program assignments.

Beyond education, our work connects to the literature on LLM reasoning (Wei et al., 2022; Nye et al., 2022). These works seek to improve end-task performance by conditioning on intermediate reasoning steps, either by training models to reason (Zelikman et al., 2022) or by sampling reasoning traces at inference time (Wang et al., 2023). Recent works have explored human-like reasoning phenomena in LLMs (Gandhi et al., 2025) such as exploring (Yao et al., 2023) and backtracking (Yang et al., 2025; Chen & Li, 2024). Also related are works showing that pretraining on synthetically generated error data can improve performance for math (Ye et al., 2025) and code (Piterbarg et al., 2025).

## 6 CONCLUSION AND FUTURE WORK

We introduce a dataset of coding traces written by real students on PENCIL CODE and find that models trained on full edit traces can more strongly learn representations of students that encode information about their coding behaviors than models trained on synthetically generated traces or final programs. Our focus on modeling students' coding behavior in an educational context is a departure from prior work on code generation, which has predominantly focused on improving task accuracy (Chen et al., 2021).

The ability to model individual student behavior from their edit traces has important educational implications. Our findings that models can predict when students will backtrack or struggle with particular concepts (Section 4.2) enable early intervention systems, such as automatically detecting whether an assignment is appropriately challenging for a student. The efficient adaptation to new students with just 2-4 traces (Section 4.4) makes this practical for real classrooms where teachers cannot manually model each learner. Furthermore, our demonstration that learned student representations can be used to recover from errors while maintaining individual coding style (Section 4.5) addresses a critical challenge in education: providing personalized feedback that respects students' learning trajectories rather than imposing a single "correct" approach. Future work can build on these findings to develop intelligent tutoring systems, automated hint generation, and assessment tools that evaluate students' learning processes rather than just final correctness.

A natural question is whether these results extend to other platforms beyond PENCIL CODE. Given the large user base of PENCIL CODE and similarity of some libraries in CoffeeScript to Python (*e.g.,* ones for turtle graphics), we hypothesize that they do but leave it to future work for empirical investigation. Finally, another interesting direction for future work is to develop alternative methods for leveraging the structural and temporal properties of edit sequences during training.

REPRODUCIBILITY

We will release the code used to train and evaluate models upon publication. We provide details on dataset preparation in §A, including on preprocessing (§A.1), anonymization (§A.2), and downsampling to match numbers of tokens (§A.3). Details on how we train models, along with hyperparameters, are in §B. Information on our general evaluation set-up is also provided in the appendix, with calculations for metrics in §D and how we sample traces from trained models in §E. §G provides details on how we adapt models to unseen students, and §H and §I give details on our probing/representational analyses. Finally, §J gives details on our error recovery experiments.

ETHICS

Given that the dataset we study contains data from school-age users of PENCIL CODE, one key area for concern is around protecting student PII. We take multiple steps to protect student data. We first obtained permission from our institution that usage of the data for research purposes is exempt under our institution's IRB. Before running all experiments in this paper, we mapped all student user IDs to randomly hashed usernames (as described in §A.1). However, for a public data release, we wanted to prevent possible leakage of PII through the text of code written by users. We therefore conducted a more thorough removal of PII in the dataset through both automatic detection methods (for example, replacing URLs, which we observed often contained personal information, with a special URL token `<URL>`) and manual analysis. Even with this thorough anonymization, a concern is that because all the code written by a student is associated with a particular ID, it might be possible to draw inferences about user identities from the code they write. We therefore decided, in an agreement with the PENCIL CODE team, to release the more fully anonymized dataset, along with a `trace` model trained on this dataset, to the broader research community through a *gated release*. This gated release will allow researchers to study the questions in this work, which require being able to associate student IDs with all of their coding traces, while also protecting and monitoring usage of this data.

More generally, while we believe the dataset collected from PENCIL CODE and associated research in the paper can lay the foundation for future work on educational tools and methods that take into account students' learning *trajectories*, any future work using this data should be done carefully to ensure safe and appropriate usage, particularly in an educational context. In particular, the continued pretraining methods we explore in this dataset use pretrained LLMs, and any biases found in the pretraining data used to train them could influence their behavior in future contexts.

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
