## A    DATASET STATISTICS

### A.1    DATA PROCESSING

**Dataset Preparation**    We received data from the PENCIL CODE team from the years 2015 to 2024. The raw data consists of HTTP GET requests to the PENCIL CODE server corresponding to code execution. With permission from the PENCIL CODE team, we only access the timestamp, a random hash of the user ID, and the current program state code and title, which is included with the request URL.[15] We construct program traces for a particular (username, title) pair by ordering all associated program code with respect to the recorded timestamp and title.

We assume that the title used by the student reflects their goal and maps onto the program semantics, as the raw data do not include other metadata about students' goals (*e.g.,* , the actual assignment they were solving). In some cases, this title may not provide sufficiently fine-grained information about a student's goal. For example, the common `untitled`, `myprogram`, and `first` titles often correspond to a diverse set of kinds of traces.[16]

**Preprocessing**    We construct traces from titles, timestamps, and programs as follows, where variables are bolded:    [**title**]<mask>...<mask><mask><start>CODE    **1** (**timestamp**):\n[**program**]\n CODE **2** (**timestamp**):\n[**program**]\n...<|endoftext|>.[17] We randomly mask 15% of trace titles during training to facilitate experiments where we evaluate models' abilities to infer trace titles from code (§4.2). We remove empty program traces and last programs for the `last` model. If consecutive programs are identical, we remove all but the last program. For traces that are longer than GPT-2's context window of 1024 tokens, we create as many inputs to cover the trace in chunks of 1024 tokens, with 64 tokens of overlap between each chunk.

Table 1: Dataset Statistics

| Split | Total Traces | Unique Students | Unique Programs |
|---|---|---|---|
| `train` | 2,941,032 | 1,110,554 | 260,428 |
| `seen student/seen title` | 507,221 | 226,673 | 62,207 |
| `seen student/unseen title` | 71,799 | 49,240 | 41,086 |
| `unseen student/seen title` | 259,345 | 208,666 | 20,073 |
| `unseen student/unseen title` | 8,814 | 6,185 | 6,558 |

Table 2: Statistics on the different dataset splits discussed in §3.2.

### A.2    REMOVING STUDENT PERSONALLY IDENTIFIABLE INFORMATION (PII)

As part of our agreement with the PENCIL CODE team to make student data publicly available, we take further steps to prevent leakage of PII about school-age children. We first filter out person names from all data splits using `nltk.corpus.names`, which consists of 5001 female names and 2943 male names, and replace them with <UNK> tokens. We then hand-examined a random set of 2000 independent log entries that contain either strings or comments with more than 10 characters total in order to identify other kinds of PII captured in the data. In this sample, we observed **347** names, though note this is a conservative estimate as some examples flagged include `cookie` and `Times New Roman`. We additionally found **112** web links that potentially contained PII, such as the flag of the student's home country. We therefore replace all URLs with a special <URL> token. Furthermore, **2** strings contained a student's school name, and **2** contained geographic location – because it is difficult to enumerate all potential locations, we instead chose a conservative approach and replaced all strings containing only 1 or 2 words with <UNK> tokens (as we observed PII only included in headers with independent strings for location). We did not observe any additional PII such as phone numbers or email addresses. Finally, we observed that there were some assignments

---

[15]This use of the data was classified as exempt by our institution's IRB.

[16]We observed that for `first` traces, users frequently appeared to switch between goals mid-trace, *e.g.,* first starting with `speed 2; pen red`, which is the provided PENCIL CODE default.

[17]There are most 50 tokens for the trace title and mask tokens.

(*e.g.,* name) where students were tasked with creating a program that graphically drew their name on the PencilCode interface. As this is a form of PII, we drop all program traces with assignment titles containing name (*i.e.,* including myname).

Removing PII from the data naturally affects the model. Therefore, all results reported in the paper use the original dataset, while we release, both all filtered data and a trace model trained on the filtered training data split.

## A.3 DOWNSAMPLING

To create the downsampled datasets for the trace and synthetic models, we continue pruning the traces in their respective training datasets until the ratio of [the difference in number of tokens in the datasets] to [the number of tokens in the last dataset] is less than or equal to 0.001.

## B TRAINING DETAILS

We continually train models with a batch size of 32 for a maximum of 3 epochs, with early stopping based on evaluations every 1000 steps (patience = 20). We use 2% of the training data as validation data during training. We train with a learning rate of $5e - 5$ with a linear learning rate scheduler and Adam optimizer. Each model was trained on an A100 GPU on an internal cluster for around 2 weeks.

## C OLMo-2 RESULTS

| Model | seen student seen title | unseen student seen title | seen student unseen title |
|---|---|---|---|
| last | $0.262 \pm 0.013$ | $0.246 \pm 0.013$ | $0.042 \pm 0.005$ |
| trace downsampled | $\mathbf{0.284} \pm 0.013$ | $\mathbf{0.276} \pm 0.013$ | $\mathbf{0.059} \pm 0.006$ |

Table 3: BLEU score evaluation results for different splits. The best-performing cell for each column is bolded. We evaluate the best-performing checkpoints for each model after up to 3 epochs of training. Results are for 100 randomly sampled titles corresponding to PENCIL CODE assignments. We sample 10 generations for each title and model.

## D METRICS

**Successful Execution** We measure whether a program successfully executes by using a headless browser to attempt to execute the student-written code. First, we append console.log("END_REACHED") to the end of the following HTML template:

```
<!doctype html>
<html>
<body><script src="https://pencilcode.net/turtlebits.js" crossorigin="anonymous"
type="text/javascript"></script><script type="text/LANGUAGENAME">

INSERT_CODE_HERE

</script></body></html>
```

We replace LANGUAGENAME with coffeescript or javascript depending on the language of the code. We replace INSERT_CODE_HERE with the student's program.

We first start with CoffeeScript language. Using a headless browser, we open the HTML file and check if the text END_REACHED is logged to the console, using at most 5 seconds for page navigation and 5 seconds for execution:

1. If no error occurs and END_REACHED is logged to the console, we consider the program to have successfully executed. We skip all other steps.

2. If an error is reached, we move to step (4).

3. If the page navigates without an error or END_REACHED being logged, it is unclear whether the program timed out because of a system delay *external* to the program causing the execution to not reach an error that otherwise would have occurred. Therefore, we retry execution with a longer timeout (adding 100 seconds to both the page navigation and execution timeouts), *unless* the code contains any of the following keywords: await, forever (with no stop()), or prompt. If one of these keywords is present, we do not retry execution, as we expect these programs to run indefinitely, and move to step (4). If after retrying with longer timeouts, there is still no error or END_REACHED logged, we move to step (4).

4. We attempt to execute the code in JavaScript instead of CoffeeScript, starting at step (1).

We run programs in both CoffeeScript and JavaScript because students could have written their code in either language on PENCIL CODE. If both languages do not lead to successful execution, we consider the execution unsuccessful. If one language leads to successful execution, we consider the execution successful.

## E  SAMPLING GENERATIONS

For the core behavioral evaluations in (§4.1), we collect 5 samples from each model for each unique student ID and title pair. For each metrics, we calculate correlation with the mean value across samples and the ground truth, except for Self-BLEU, for which we report the score across the 5 samples.

Because full student program traces often exceed limit of 1024 output tokens, we repeatedly generate 3 times, stopping sooner if an |endoftext| token is generated. If the last program of the generated text does not contain |endoftext|, we then treat the preceding program as the final state.

## F  BEHAVIORAL RESULTS WITH FILTERING

See Figure 9.

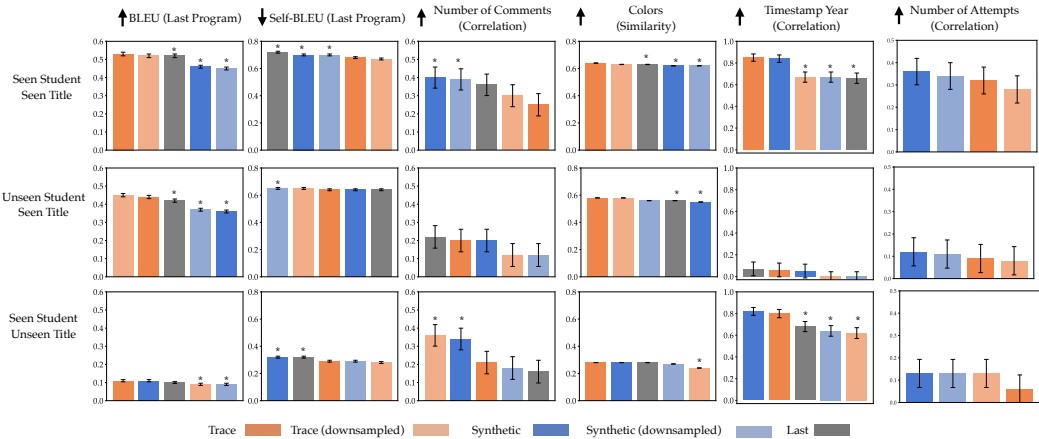

Figure 9: **Correlation of Generated Trace Properties with Ground Truth (Final Program State)** We evaluate the generated final program state of a trace from sampling all models across evaluation splits. Correlation denotes Pearson's correlation coefficient. The Colors metric compares the cosine similarity of between program color embeddings. * indicates a statistically significant difference with the trace model using a paired T-test between unique (student, title) pairs at $p = 0.05$ with Bonferroni correction, and error bars indicate standard errors of the mean.

# G    ADAPTING TO NEW STUDENTS

We finetune the `trace` model for up to 100 epochs with early stopping patience of 10. We use a batch size of 32 and learning rate of 5e-4 with no masking of program names. We freeze all model weights except the parameters of the student embedding MLP. We keep the bias terms of the MLP frozen.

We use 5% of the unseen students for hyperparameter selection to determine the optimal number of training epochs. We first train on the 5% hyperparameter selection set with early stopping, identify the best epoch, and then train the remaining 95% of students for that number of epochs.

We generate rollouts for a sample of 100 of the students whose traces are finetuned on. We filter degenerate traces, *i.e.,* those that do not have any matched programs, then evaluate on the intersecting subset of held out data for all values of $k$ after filtering. At most 0.01% of generated traces are filtered for any given $k$.

# H    PROBING STUDENT EMBEDDINGS

To construct student embeddings, we probe the first layer of the student embedding module (a 2-layer MLP with embedding dimension = 768 and hidden dimension = 64).

The probe models are MLP regressors with 2 hidden layers, each with size 100. We train the probes for a maximum of 5,000 iterations with learning rate 0.001, batch size 64, and early stopping.

# I    PROBING CODE EMBEDDINGS

We present details on our experiments probing code embeddings (§4.2).

**Constructing Datasets for Probes**    Let a trace for student $s$ and assignment $a$ be denoted by $\tau = (p_0, p_1, \ldots, p_T)$, where $p_0$ is the initial empty program (containing only the `CODE` prefix) and $p_T$ is the final program. For each $\tau$, we consider all prefixes up to each time step $t$, *i.e.,* , for $t = 0, 1, \ldots, T$, the prefix $\tau_{:t} = (p_0, p_1, \ldots, p_t)$. For each prefix $\tau_{:t}$, we construct an input $x = (\tau_{:t}, s, a)$, consisting of the programs in the prefix, the student ID, and the trace title. We then obtain a code embedding $e = f_{\text{trace}}(x)$ by conditioning the `trace` model on $x$ and taking the mean of the token embeddings at the last layer.[18] For each property, we construct a dataset $\mathcal{D} = \{(e^j, y^j)\}_{j=1}^N$ where $e^j$ is the embedding for input $x^j$, and $y^j$ is the property to be predicted. We exclude any $x$ whose tokenized length exceeds `GPT-2`'s context window.

**Filter Probing Datasets**    We filter the probing datasets to remove confounders in the metrics. For example, to predict whether the student will deviate from their goal (the final program), we only include prefixes that do not contain the final program in the trace, *i.e.,* $D = \{x^j : \tau_{:t}^j \neq \tau_{:T}^j\}$. For probing trace title, we set $D = \{x^j : \tau_{:t}^j \neq \tau_{:0}^j\}$, *i.e.,* we only include inputs that contain some code (excluding the initial empty program). For predicting whether the program is the final program, we do not filter $\mathcal{D}$ (we include all inputs). For all other metrics, we only include inputs that do not contain the final trace. The reasons are twofold: First, we wish to avoid possible leakage from the final program (*e.g.,* predicting whether the final program will be correct is not a future-looking metric if the final program is given in the input). Second, because many of the metrics correlate with whether the program is the final program (*e.g.,* the edit distance to the final program is 0 if the program is the final program), a probe may rely on information in the input about whether the program is final and make its predictions based on this information, rather than making sophisticated inferences about the metric directly; given that we already have a metric directly predicting whether the program is the final program, we remove this confounder in probing for other metrics by filtering $\mathcal{D}$ to only include inputs that do not contain the final program in the trace, *i.e.,* $\{x^j : \tau_{:t}^j \neq \tau_{:T}^j\}$.

**Probes**    We train a single probe on the filtered dataset for each metric. The probes are ridge regressors and classifiers, implemented using the `scikit-learn` `RidgeCV` and `RidgeClassifierCV`

---

[18]See §I for justification using Centered Kernel Alignment (Kornblith et al., 2019).

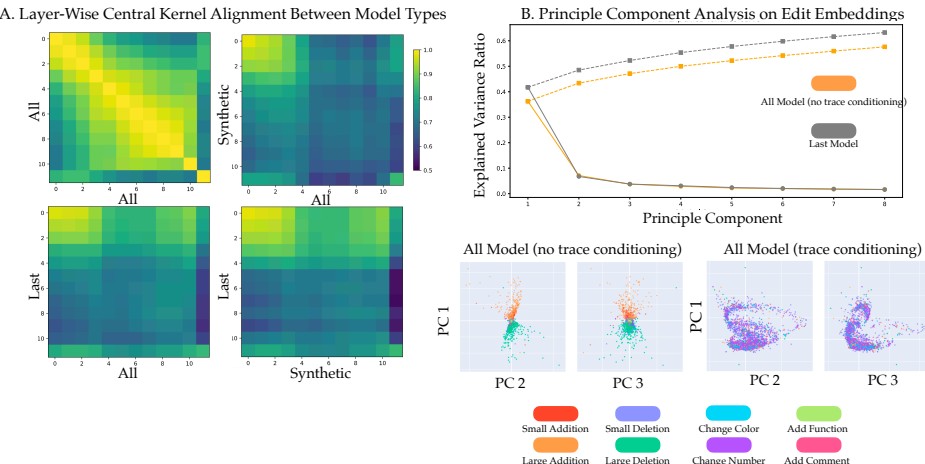

Figure 10: (Left) Running Central Kernel Alignment Kornblith et al. (2019) on on code embeddings across layers of the `trace`, `synthetic`, and `last` models shows that the three models differ significantly in the last embedding layer, with the `last` model causing asymmetry. (Right) Running PCA on the differences between two consecutive program states in a trace results shows that embeddings encode information relevant to the edit type (e.g. small addition).

classes. We perform cross-validation to pick the value of $\alpha$, *i.e.,* the regularization parameter, from the set of values $\alpha \in \{1, 1e-1, 1e-2, 1e-3, 1e-4, 1e-5, 1e-6, 1e-7\}$.

**Embedding Analysis**    Finally, we take a closer look at the learned embeddings across the `trace`, `last`, and `synthetic` models to better understand their representational differences. We first measure similiarity between embeddings across all layers for each pair of modes, and find strongest differences in the last layer, particularly for the `last` model trained only on final program states (Figure 10). When running Principle Component Analysis (with 8 components) on the `trace` and `last` model, the first two principle components explain more variance in the data for the `last` model, suggesting that the `trace` model learns more complex representations. Finally, we observe that the first two principle components are well aligned with student edit types (*e.g.,* small addition), though conditioning on an entire trace leads to more complex structure.

## J    MODEL STEERING

We showed in Section §4.5 that we can improve assisted error recovery with the `trace` model by using a "strong student embedding" instead of the original. To find this embedding, we sort all students in our training dataset by the average goal backtracking metric across their program traces. We then selected the student with the lowest degree of backtracking who had completed more than 20 different programs. Future work can identify other attributes for model steering via student embeddings, such as the language used in any text (*e.g.,* Greek), certain colors, or high speeds for interactive programs.

We additionally trained a more complex `synthetic-complex` model where the synthetic edits can either add or delete any number of lines from any location in the current program state. Therefore, any two arbitrary program states can be connected with a sequence of edits. We find that, for error recovery, the "trace" model still outperforms the `synthetic-complex` model, which only slightly outperforms the `synthetic` model at lower values for time (x-axis, see Figure 8). Results for `synthetic-complex` include the following values:

On a qualitative level, we believe the reason for this poor performance is that the method for creating `synthetic-complex` treats all lines equally, and is not able to capture how some aspects of a program (*e.g.,* a function definition) are more susceptible to incorrect implementations than others, hindering error recovery.

| Time (s) | Trace % Correct | Synthetic % Correct | Synthetic-Complex % Correct |
|---|---|---|---|
| 0.01 | 0.64 | 0.39 | 0.41 |
| 1 | 0.60 | 0.39 | 0.40 |
| 60 | 0.64 | 0.45 | 0.40 |

Table 4: Performance across different time scales.

# K  EXAMPLE PENCIL CODE ASSIGNMENTS

PENCIL CODE assignments vary in terms of complexity. Here we provide examples of two common assignments in our dataset with a moderate degree of complexity, showing that our dataset extends beyond simple introductory tasks.

**Riddle Assignment**

```
# This function looks at the words in your
# answer and picks the troll's response.
replyto = (a) ->
  for word in a.toLowerCase().split /\W+/
    switch word
      when "secret"
        return "Right."
      when "money"
        return "I don't want your money!"
      when "advice"
        return "My advice: think again."
      when "who"
        return "I am the Bridge Troll!"
      when "troll"
        return "The troll laughs at you."
      when "know"
        return "I know the answer. Do you?"
  # If no words were recognized, a random
  # canned response is picked.
  return random [
    "That doesn't make sense."
    "Guess again!"
    "That is not right!"
  ]

riddle = ->
  write """
    A troll blocks the bridge and asks:
  """
  type """
    When you don't have me,
    you want me,
    but when you do have me,
    you want to give me away.
    What am I?
  """
  while s isnt "Right."
    # Wait for an answer and reply to it.
    await readstr defer a
    s = replyto a
    write s
  do proceed

proceed = ->
```

```
write """
  The troll steps aside, and you cross.
  (what happens next?)
"""

# start the game
do riddle
```

**Rhythm Assignment**

```
  clapsound =
wave: 'noise'
attack: .01
decay: .1
gain: .5
cutfollow: 4
resonance: 10

lowdrum =
  wave: 'noise'
  attack: .06
  decay: .15
  gain: 10
  cutfollow: 0.7
  resonance: 5

blocksound =
  wave: 'noise'
  attack: .05
  decay: .1
  gain: 0.3
  cutfollow: 3
  resonance: 5

clap = img 'percussion-clap', 100
block = img 'percussion-woodblock', 100
drum = img 'percussion-timpani', 100
sync clap, block, drum

for i in [1..3]
  clap.play
    timbre: clapsound
    tempo: 100
    song: "zc/c/ccc/c/c/c//c/c//cz/"

for i in [1..3]
  block.play
    timbre: blocksound
    tempo: 100
    song: "z/G/GGGz/G/GGG"

for i in [1..3]
  drum.play
    timbre: lowdrum
    tempo: 100
    song: "DDDDDDDD"
```