# OpenReview forum: "Modeling Student Learning with 3.8 Million Program Traces"
_ICLR.cc/2026/Conference — Submitted to ICLR 2026_

### Official Review · Reviewer_ncN7 · 2025-10-26

**Soundness:** 3
**Presentation:** 3
**Contribution:** 3
**Rating:** 6
**Confidence:** 4

**Summary:**

This paper introduces a dataset of over 3.8 million programming reasoning traces from a free online educational platform. The authors develop and compare five model variants trained on this dataset: the trace model, last model, synthetic model, trace downsampled model, and synthetic downsampled model, which are evaluated from both behavioral and representational perspectives. They demonstrate that models trained on full traces acquire stronger representations of student coding behavior compared to models trained solely on synthetically generated traces or final program submissions.

**Strengths:**

1. This paper is well-motivated, and a decent amount of technical details are given.
2. The idea of modeling students' coding behavior through intermediate traces is both interesting and practical.

**Weaknesses:**

1. Insufficient dataset presentation
2. Missing discussion of related work and evaluation metric
3. Lacks user study
4. Limited evaluation to outdated model (GPT-2)
5. Code not provided

**Questions:**

1\. **Concerns about the dataset presentation**

A key contribution of this paper is the presented programming reasoning traces dataset. I suggest the authors add a dedicated section within the main text to thoroughly introduce the dataset's features and characteristics rather than placing this important information in the appendix. Additionally, providing an illustrative visualization of the dataset structure would help readers better grasp its organization and content.

2\. **Missing discussion of related work and evaluation metrics**

For the behavioral evaluation, the authors compare generated samples against actual student-written code. This objective semms similar to the work "Open-ended Knowledge Tracing for Computer Science Education" (EMNLP, 2022), which should be cited and discussed. Also, I suggest adopting CodeBLEU—a variant of the traditional BLEU metric specifically adapted for code—as suggested by this related work, as it would allow for a more accurate assessment of similarity between the predicted and actual student code.

3\. **User study**

The authors demonstrate that their trace model can help students recover from errors. I suggest that the authors conduct a user study in real educational settings to further validate this claim. Such an evaluation would provide valuable empirical evidence for the practical effectiveness of the proposed model.

4\. **Clarification on Figure 6 results**

In Figure 6, as the number of fine-tuning traces increases, the performance on trace generation appears to be lower compared to final program generation. Could the authors provide a more detailed analysis or explanation of this phenomenon?

5\. **Evaluation on more advanced language models**

The authors conduct experiments using base GPT-2 and OLMo-2 models. Given that GPT-2 is somewhat outdated, I suggest extending the evaluation to include more advanced models, such as those from the Llama series or other state-of-the-art LLMs, to further strengthen the generalizability of the findings.

6\. **Code and reproducibility**

The authors are encouraged to release the code to facilitate reproducibility and benefit the research community.

7\. **Typo**

Page 4, line 202: "a an" → "an"

**Details Of Ethics Concerns:**

The authors have adequately discussed the protection of student information in the manuscript, and there appear to be no ethical concerns.

---

> ### Author Response · Authors · 2025-11-19
> **Overall Comment**
>
> Thank you for the thorough review, feedback, and questions!  We are glad to hear that the reviewer found the paper to be **well-motivated** and the key ideas to be **interesting.** We have addressed weaknesses below. Please let us know if there are any further questions we can address!
>
> 1. **Insufficient dataset presentation**: Thank you for raising this point! We agree that more thorough presentation of the dataset would strengthen the paper, and we have moved from the Appendix to the main text a dedicated paragraph with dataset statistics and features in the revised paper, noting that they were originally placed in the Appendix due to space constraints.
>
> 2. **Related Work and CodeBLEU**: Thank you for the reference – we agree it is relevant  to cite and have updated the paper accordingly. Regarding CodeBLEU, we did in fact implement it, and found that BLEU was more discriminative between different Pencil Code programs. We hypothesize that because Pencil Code programs tend to have a linear flow of imperative commands, CodeBLEU might inappropriately overweight certain keywords, and expect AST match and rich structure. We believed it was important to proceed with the more discriminative metric, and have added a footnote to mention this.
>
> 3. **User Study**: While we agree a user study would be interesting, conducting rigorous educational studies is non-trivial and we believe goes beyond this paper's scope. In addition to requiring a separate IRB approval due to working with K-12 students, any interventions would need to be integrated with the Pencil Code platform, which opens up several UI questions. We believe a good user study would appropriately control for UI design, which we view as orthogonal to our paper's contributions, which establishes the technical foundations for any future deployment. We hope that our findings can inform development of systems that can be studied in future user studies.
>
> 4. **Figure 6**: In Figure 6, trace BLEU is lower than final program BLEU because evaluating full traces is harder: the model must match the entire temporal sequence, not just the final state. Additionally, with limited fine-tuning data (k=2-4 traces), the model is likely to learn the student's final goals more quickly (especially for assignments it has seen other students' solutions for), rather than their complete editing process. We hope this helps explain the gap!
>
> 5. **Advanced Language Models**: While we acknowledge GPT-2 is an older model, our research question is about training data (traces vs. synthetic vs. final-only), not achieving SOTA performance. We believe controlled experiments with smaller, well-understood models are scientifically appropriate. That said, in Table 3 in the Appendix, we provide results for the OLMo-2 model (1B).  We show not only that our findings hold across scale, but also that the gap between trace and last models widens with increasing model size (i.e. for OLMo-2).
>
> 6. **Code and Reproducibility**: Yes, we intend to release our code, trained models, and the anonymized dataset (via gated release for privacy protection) upon acceptance to facilitate reproducibility.
>
> 7. **Typo**: Thank you for pointing this out! We have fixed this in the revision.

---

### Official Review · Reviewer_ngbA · 2025-10-29

**Soundness:** 2
**Presentation:** 2
**Contribution:** 2
**Rating:** 2
**Confidence:** 5

**Summary:**

This paper introduces a 3.8M programming trace dataset from Pencil Code and trains language models to capture student coding behavior, comparing models trained on real traces, synthetic traces, and final programs only. The focus on modeling "how" students code rather than just "what" they produce is interesting, but the scope is limited and the experimental section needs significant reorganization.

**Strengths:**

* The focus on "how" students code instead of just "what" they produce is a valuable perspective shift for modeling programming behavior.
* The 3.8M trace dataset from real students over 9 years is substantial and could benefit the education and code generation community.

**Weaknesses:**

* The base models (GPT-2 124M, OLMo-2 1B) are outdated. Modern models (such as qwen3, starcoder) would be more convincing baselines.
* Line 132 mentions "reported in Table 3" but I cannot find Table 3 anywhere in the paper.
* The entire work is based on one platform (Pencil Code) teaching "simple programming concepts" with visual blocks. This feels too narrow for ICLR. There is no evidence the findings generalize to other languages, platforms, or more complex programming tasks.
* The citation format does not follow ICLR style. Please check the formatting guidelines.
* Figure 3 has overlapping numbers that make it hard to read. Please fix the visualization.
* The experimental results section is very hard to follow. There are too many sub-research questions (5 major sections, each with multiple questions) but they are not well-justified. For example, Section 4.1 asks "Can models generate code that reflects real student programming behavior?" but I don't understand why this matters. The model is still just generating programs, so what is the point? The later experiments on probing and adapting to new students are more interesting, but they get buried.
* The paper tries to answer too many research questions at once. The authors should narrow down to 2-3 core questions and go deeper on those instead of spreading thin across many shallow analyses. More research questions does not equal a better paper.

**Questions:**

see weakness above

---

> ### Author Response · Authors · 2025-11-19
> **Overall Comment**
>
> Thank you for the review and helpful feedback! We believe there are some important misconceptions surrounding our work, which we address in separate comments below. Please let us know if there are further questions we can address.

---

> > ### Author Response · Authors · 2025-11-19
> > **Comment 1: Baselines + Table 3**
> >
> > We respectfully disagree that using GPT-2 and OLMo-2 undermines our contribution, as this paper’s goal is **not** to achieve **state-of-the-art code generation**. Instead, our research question is fundamentally about **the training data and objectives used to train any base model** – i.e. whether models trained on real student edit traces learn different representations than models trained on synthetic traces or final programs alone. Concurrent work by [Miroyan et al., 2025](https://arxiv.org/pdf/2507.12674) has also shown that finetuning models, including Qwen-family models, on student code trajectories significantly improves alignment with real student code trajectories, highlighting that **finetuning/continued pretraining on student data is key** for asking the kinds of questions central to our analyses about what can be learned about student behavior and code.
> >
> > We focus on GPT-2 (124M) and OLMo-2 (1B) for this work due to practical advantages, as they are accessible and trainable with modest resources. Our results show not only that our findings hold across scale, but also that the **gap between trace and last models widens with increasing model size** (i.e. for OLMo-2 vs GPT-2). We therefore expect that training more powerful/larger models on the dataset would make training on traces even more valuable, and we hope future work can build on our findings and dataset to test this hypothesis.
> >
> > ***Note**: Table 3 is in the Appendix, which was uploaded as Supplementary Materials. We have modified this line to say "Table 3 in the Appendix".
> >
> > For the reasons discussed above – i.e. that our contribution is not to introduce a method for improving code generation, but to ask what can be learned from such data – there are no real existing “baselines”. That said, in light of the reviewer’s question about how modern models like StarCoder would perform at some of the tasks we explore in the paper,  we ran **new experiments with a ~15B pretrained StarCoderBase model** to qualitatively evaluate whether it can model student code through ICL. We prompted this model with few-shot examples to facilitate in-context learning and found that the model ends up copying from context rather than producing meaningful generalizations in three settings we tested:
> >
> > 1. **Cannot generalize to new program names**: When given a program name not included as a few-shot example, the model cannot generalize to write code for the new program name. For example, in both **examples A and B** below, the model outputs variants of the few shot example for program name colors for the new program names *house* and *star*.
> >
> > 2. **Cannot make meaningful trace edits**: We also experimented with fixing a program name (i.e. to star) and giving StarCoderBase few-shot examples of different traces for that program name, then prompting it to complete the coding trace given a real student’s first program in another *star* trace. Again, we find in both **examples C and D** below that the model only changes timestamps, repeating code from the first program in the trace (rather than making edits to the code).
> >
> > 3. **Cannot recover from errors**: We also experiment with giving the pretrained StarCoder model a trace with a program that has an error to see whether the model can use ICL to recover from the error. (We few-shot prompt with examples of error recovery). As shown in **example E**, we find that the model cannot fix the bug in the given trace and instead again repeats the given code. As shown in **example F**, even with GPT-5, a powerful closed-weights model, we find that the model cannot fix the bug in the trace, and even generates an unrelated program within the trace.
> >
> > (Examples are included in separate comments below.)

---

> > > ### Author Response · Authors · 2025-11-19
> > > **Comment 2: PencilCode Scope**
> > >
> > > While we acknowledge our work is limited to one platform, we believe the contribution remains significant because
> > >
> > > (1) to the best of our knowledge, this is the **first large-scale dataset of real student programming traces** suitable for LLM training that also has high density (many students solve the same assignment, and each student solves many assignments),
> > >
> > > (2) Pencil Code supports **diverse activities** (turtle graphics, music, networking, storytelling) across multiple languages (CoffeeScript, JavaScript, HTML, CSS), providing substantial within-platform variety, and
> > >
> > > (3) our approach of comparing trace-based vs. synthetic vs. last-state-only training is broadly applicable.
> > >
> > > Furthermore, our methods (steering edit recovery via student embeddings and comparing trace-trained vs. final-program-trained models) are **language-agnostic**.
> > >
> > > We have updated the Appendix to show examples of complex assignments on Pencil Code that are similar to standard programming assignments in other datasets / platforms.  Finally, even if interventions based on our work were limited to PencilCode, deployment would reach a diverse range of **over 1M students**, which we believe is meaningful.

---

> > > > ### Author Response · Authors · 2025-11-19
> > > > **Comment 3: Experimental Results Section Hard to Follow**
> > > >
> > > > Thank you for your feedback on making the experimental results section more clear!
> > > >
> > > > We believe there is an important potential misunderstanding around the purpose of Section 4.1. The research question "can models generate code that reflects real student programming behavior?" is not asking simply whether models can generate syntactically correct programs. Instead, we are asking whether they capture the **temporal dynamics** and **exploratory patterns** characteristic of how real students learn to program. Specifically, we show that **trace**-trained models generate sequences with **realistic goal backtracking, edit patterns, and diversity** that match ground truth student behavior, while the synthetic and last-only baselines don't capture these dynamics as strongly.
> > > >
> > > > This behavioral validation is important: if models trained on traces were simply memorizing syntax without learning meaningful patterns about the programming process, all subsequent claims about learned representations (4.2-4.3) and educational applications (4.5) would lack any grounding.
> > > >
> > > > We have **revised the start of the Experiment section** to explicitly state why behavioral validation precedes representational analysis (i.e., probing), and include the logical progression of our experiments:
> > > >
> > > > (1) first validate that models behaviorally capture student learning dynamics
> > > >
> > > > (2) probe what information is encoded in representations that explains such behavioral differences
> > > >
> > > > (3) demonstrate how to use trace models to personalize to new students and intervene for error recovery

---

> ### Author Response · Authors · 2025-11-19
> **Comment 4: Too many research questions**
>
> To the best of our knowledge, our work is the first to train language models on large-scale real student edit traces. This opens up several questions about what these models learn, which we explore deeply via different experiments.
>
> The core contributions from our analyses can be summarized as three main findings:
>
>  (1) models trained on real traces learn **richer representations** than those trained on synthetic traces or final programs only—demonstrated through both **behavioral generation** (4.1) and **representational probing** (§.2-4.3)
>
>   (2) these representations encode **student-specific information** that enables **efficient personalization** with just 2-4 examples (4.4); and
>
>   (3) learned representations enable practical educational applications like style-preserving **error recovery** (4.5).
>
> Each experimental section provides evidence for one of these three questions with multiple **controls** (shuffled embeddings, downsampled datasets, filtered analyses), so this is not shallow exploration: we go deep on each question while maintaining breadth.
>
> **We hope that the revised introduction in Section 4, which organizes findings into three key contributions, has addressed the reviewer’s concern. If not, we would greatly appreciate suggestions for what structure would be more compelling.**

---

> > ### Author Response · Authors · 2025-11-19
> > **Comment 5: Citation format + Figure 3**
> >
> > Thank you for the feedback! However, we do not fully understand what aspect of Figure 3 shows overlapping numbers. Could you please clarify?
> >
> > Furthermore, to the best of our knowledge, colored URLs for citations are not against ICLR formatting guidelines. (See Section 4.1 in the style file, which just requires that the citations are alphabetically listed: “As to the format of the references themselves, any style is acceptable as long as it is used consistently.”) Could you please clarify your concern?

---

### Official Review · Reviewer_fQu1 · 2025-10-29

**Soundness:** 2
**Presentation:** 2
**Contribution:** 2
**Rating:** 2
**Confidence:** 4

**Summary:**

This work involves a set of model training experiments on a large programming traces dataset from Pencil Code. Specifically, they used five models to model students' behaviors, and also investigated the representations of both code and students.

**Strengths:**

+ It is a major contribution to show that, at a large scale, programming traces are useful for modeling students' programming.

**Weaknesses:**

- The contribution of this paper is unclear. One key issue is that while it comes from an educational discipline, it does not have any task involving actual educational goals. The five models are all about student behaviors or representations of students or code, but what is the next step? There is almost no educational implication discussed in the work.
- There are some key claims counterintuitive for general machine learning tasks. One of the biggest issues is about student embedding -- how exactly can we expect a model learned with student IDs to be generalizable for future new students? There are discussions about the result when new students are involved, and the result says "generalization is still difficult" -- this is almost certain, even for a large language model now trained with a lot of data. If you ask GPT-4 who a student is, it likely won't give you any good idea. The power of generalizability cannot help with tasks like overfitting to IDs.

**Questions:**

- Line 89: Why is this large dataset suitable for language model training? For small language models, smaller datasets could also work, especially if we want to create models for specific contexts.
- Line 94: The requirement and context of learning will be very important for the final program states. In classroom settings, students' final program states are almost all correct, while in situations of informal learning, there's often a lack of motivation for students to finish programming for many. This is actually not a minor issue -- context is very important for educational applications and this is missing.
- Line 99: We cannot train from IDs, but can check about certain classes or sessions.
- Line 118: So what is the goal? Education happens in a certain context, and it will need to show it surpasses small models in their own context to make sense. Otherwise we can always use smaller models trained in specific contexts.
- Line 130: While training LMs are important, it is still important to show what exactly will be a good downstream educational task.
- Line 140: I don't get this -- student IDs are involved in training and in this case, any new student will be an unknown input.

---

> ### Author Response · Authors · 2025-11-18
> **Overall Comment**
>
> Thank you very much for the thorough review and thoughtful feedback/questions! We believe there are some misconceptions surrounding the paper’s contribution and role of student embeddings, which we address in separate comments below. Please let us know if there are further questions we can address.

---

> ### Author Response · Authors · 2025-11-18
> **Student Embedding Generalization**
>
> We believe there may be a misunderstanding about how we build student embeddings. Our claim is not that models learn to generalize to unseen students in the sense of predicting properties of completely new individuals from their IDs alone. As the reviewer notes, that would be impossible.
>
> Instead, our work shows that (1) **student-specific behaviors** can be learned from edit traces and **encoded in embeddings** (4.3), (2) the information in these embeddings is **useful for predicting future student behavior**--both across traces (4.1) and within a single trace (4.2)--and (3) these embeddings can be efficiently **adapted to new students** using parameter-efficient fine-tuning with only a **few examples** (4.4).
>
> Our approach is **analogous to few-shot personalization in other domains**: the base model learns what kinds of individual variation exist (e.g., some students comment frequently, others backtrack often), and then **model adaptation quickly identifies where a new student falls** along these dimensions.
>
> Finally, the "generalization is still difficult" comment in Section 4.1 actually refers to generating code for unseen assignment titles (a task requiring semantic understanding of novel prompts), not to learning about new students given their traces.
>
> We have **updated the paper in 4.4** to clarify this distinction and emphasize that student embeddings are a mechanism for efficient personalization, not for zero-shot inference about unknown individuals.

---

> > ### Author Response · Authors · 2025-11-18
> > **Educational Goals**
> >
> > We respectfully disagree with the reviewer that our paper lacks connection with education implications and clarify the connections in this comment.
> >
> > We see our primary contribution as asking **what can be learned about individual students and their future behavior** from a large-scale dataset of student reasoning/coding traces. Being able to model student behavior is a **necessary requirement** for building systems that are useful in downstream educational scenarios. To that end, our findings that models can **predict when students will backtrack** from their goals or struggle with particular concepts (4.2) enable **early intervention systems** (e.g. automatically detecting whether an exam is of appropriate difficulty for the student). The ability to **efficiently adapt to new students** with just 2-4 traces (4.4) makes this **practical for real classrooms** where teachers cannot manually model each student.
> >
> > We hope that future work can build on our findings to explore their downstream applications. As an exploratory step in this direction, Section 4.5 demonstrates one potential application of our findings: we use learned student representations to **recover from errors** in real Pencil Code student’s programs while maintaining their individual coding style. Importantly, we show that this ability is **steerable with trace models**: modifying properties such as the time before the next step or the student embedding affects the model’s subsequent edits. This is **useful for feedback generation** (see [Piech et. al., 2015](https://proceedings.mlr.press/v37/piech15.html), which was published in ICML) and addresses a critical challenge in education: providing personalized feedback that respects students' learning trajectories rather than imposing a single "correct" approach.
> >
> > We appreciate the questions around educational implications and we have **expanded Section 6 to more explicitly discuss the educational implications** of our work. These include possible integration with intelligent tutoring systems, automated hint generation, and assessment tools that evaluate the student’s learning process rather than just final correctness.

---

> > > ### Author Response · Authors · 2025-11-18
> > > **Questions**
> > >
> > > We address your questions below:
> > >
> > > 1. **Line 89 (Dataset size)**: We agree that smaller datasets work for small models in specific contexts. However, our dataset's scale (3.8M traces from 1M+ students) is valuable because it captures diverse student behaviors across a large set of assignments. This diversity is crucial for learning generalizable patterns about how students code, as students differ on how they approach problems, make mistakes, and recover from errors.
> > >
> > > 2. **Line 94 (Context)**: Thank you for raising this excellent point! We agree about the importance of learning context, and Pencil Code indeed spans both formal classroom use and informal exploration. However, assignment names can often reflect which setting the student is in (e.g. “untitled” vs. “hw1:snowman”), and student embeddings can also capture whether the student often engages in exploratory behavior. In fact, Figure 4 shows that student embeddings improve accuracy when predicting whether a program state is half-way or the end of an assignment, suggesting that student embeddings capture information about the trace trajectory.
> > >
> > > We believe that the importance of learning context actually motivates our work: real-world learning environments include both structured assignments and open-ended exploration, and a good student model should be able to capture both.
> > >
> > > 3. **Line 99 (Training from IDs)**:  Our primary use of student IDs is as a mechanism to group traces from the same individual and learn personalized embeddings, and our work supports different levels of granularity. For example, trace models learn "this student tends to backtrack frequently" rather than memorizing specific identity information. In deployment, these could indeed be replaced with session IDs, class cohorts, or skill-level indicators. Since unseen students are initialized with random embeddings, one can easily assign the same ID to a group of students. In general, our key insight is that modeling individual variation (whatever the grouping) improves predictions about future behavior.
> > >
> > > 4. **Line 118 (Goal and context-specific models)**: Our goal is not to claim that one universal model beats all context-specific models, but rather to demonstrate that training on edit traces (versus just final programs) captures richer information about programming behavior. The comparison is *within* our dataset: trace models outperform models trained on the same data but without past edit history. Whether to use our model or train smaller context-specific models is a deployment decision that depends on the available resources. Our work establishes that edit traces are valuable training signals for a large set of students and assignments, which is a finding that applies whether you're training a small classroom-specific model or a larger general one.
> > >
> > > 5. **Line 130 (Downstream educational task)**: As described in our other comment, we do indeed demonstrate error recovery (4.5) as a concrete downstream task: helping students fix broken code while maintaining their style. Beyond this, our learned representations enable several educational applications: (1) early warning systems that identify when students are struggling (using predictions of backtracking and time-to-completion from 4.2), (2) personalized hint generation that adapts to individual learning patterns, and (3) student assessments that evaluate learning process and full understanding rather than just  correctness. We have expanded the discussion to make these applications more explicit.
> > >
> > > 6. **Line 140 (Student IDs and new students)**: This is an important point for clarification. During training, the model learns a student embedding layer that maps IDs to representation vectors. These learned embeddings essentially represent what dimensions of variation exist across students (commenting style, backtracking frequency, time spent, etc.). For a new student, we do not expect the model to be able to infer information about the student from their ID alone, without having ever seen any code written by the student. Instead, we ask in Section 4.4 how many coding traces the model needs to see from a student in order to learn meaningful information about the student’s behavior: We find that we can efficiently adapt the pretrained models by fine-tuning just the embedding layer on 2-4 traces from the new student. We have added a line clarifying that new student IDs are mapped to random embeddings that can be updated as the student progresses, akin to related research on model personalization.

---

> > > > ### Author Response · Authors · 2025-11-18
> > > > **Any remaining questions?**
> > > >
> > > > We hope our comments have addressed the reviewer’s key concerns and questions. Please let us know if we can elaborate on any remaining questions. Thank you for your feedback!

---

> > > ### Comment · Reviewer_fQu1 · 2025-11-24
> > >
> > > Thanks for the willingness of adding educational implications in the paper -- I will look forward to reading it in the future.

---

> > ### Comment · Reviewer_fQu1 · 2025-11-24
> >
> > Thanks for the reply -- I read through 4.1 and still found it a bit hard to accept this idea. You can use observations to embed students, and this can be generalized. But for student IDs, they are nominal data, and semanticall,y they don't hold information except to identify students. You can for sure identify the actual year of students, but how about directly using the year information? However, what about a new year, like in 2027, if we want to use the model, then we don't have any observations in 2027, and this information won't be useful.

---

> > > ### Author Response · Authors · 2025-11-27
> > > **Student IDs**
> > >
> > > Thank you for the continued engagement! Experiments involving students embeddings are actually in 4.3 and 4.4, not just in 4.1. There may still be some confusion about how student IDs function in our model, which we hope to clarify below.
> > >
> > > **1. Student IDs are not features, but are indices for learned embeddings:**
> > > Student IDs in our model work similar to word tokens in language models. The ID itself carries no semantic information, but it serves as an index to look up a learned embedding vector that captures behavioral patterns observed in that student's traces.
> > >
> > > When we say "the model learns student embeddings," we mean that during training on student A's traces, the model adjusts student A's embedding vector to capture patterns like "this student writes concise code" or "this student backtracks frequently when encountering loops".
> > >
> > > These embeddings don't encode the ID "A" itself, but rather behavioral tendencies that the model has observed during training
> > >
> > > 2. **New students (the 2027 example)**
> > > The reviewer is definitely right that a completely new student in 2027 with zero traces cannot benefit from any personalization with the model due to an ID alone. This is similar to a new user to AI chat assistants like ChatGPT not benefitting from its "memory" feature.
> > >
> > > This is exactly why our Section 4.4 focuses on few-shot adaptation: after 2-4 traces, a student's random embedding is updated to capture meaningful information. The pretrained model is still valuable as it already knows what dimensions of variation exist (from 1M+ training students), so it quickly identifies where this new student falls along those dimensions.
> > >
> > >
> > > 3. **Why not just use year information directly (like 2027)?**
> > > Year is only one coarse feature, and we believe from experiments that student embeddings capture a lot more information, such as backtracking or number of comments. That said, during deployment one could create clusters from student embeddings and create more coarse IDs. Our key insight is that modeling individual variation (however you group individuals) improves predictions, and that this variation can be efficiently learned from just a few examples.
> > >
> > >
> > > We hope this helps address any confusion!

---

### Official Review · Reviewer_s2Vq · 2025-10-31

**Soundness:** 2
**Presentation:** 3
**Contribution:** 2
**Rating:** 4
**Confidence:** 5

**Summary:**

The paper presents an analysis of a dataset of 3.8 million code editing traces.
These traces are taken from PencilCode, which is a web-based code editing
platform focused on education. PencilCode allows the user to read and edit code
in both textual and graphical form, and seamlessly switch between the two.
However, this paper focuses on the textual representation.

The paper performs continuous pre-training or fine-tuning (you can argue which)
of GPT2-124M and Olmo-1B using the trace dataset. Each training item is a
sequence of code edits, along with certain metadata such as student ID. The
paper ablates the training data format: using synthetic traces (assuming each
step adds a line), using the ground-truth traces, and using just the final
program. The natural traces perform best on several days. The tasks considered
include getting the trained models to correct errors in student traces (i.e.,
completing a student trace to be correct), predicting the program title from the
trace, etc.

**Strengths:**

- This paper presents a dataset that is potentially very interesting. However, I
  believe there is no plan to release the dataset publicly.

**Weaknesses:**

The primary weakness of this paper is that it is missing several obvious
baselines that involve prompting pretrained models (e.g., any open-weight model
that is 32B+ or even a proprietary model). Since the traces involve program
execution in JavaScript and CoffeeScript, I imagine that a reasonable pretrained
model will pick-up enough in-context signals given the trace and a reasonable
prompt. I thought the most interesting task in the paper was on L428, where the
fine-tuned model completes a prefix of a student-written trace that ended in
failure with a successful trace 60% of the time. I expect that if you give a
broken program or trace to a reasonable pretrained model, it will identify and
fix the bug at least as well. I don't expect a pretrained model to be good at
probing student representations, but it's worth asking if they can do the other
code representation tasks. E.g., asking "will a student backtrack" is similar to
asking "is there a bug".

I also think this paper needs to do a better job engaging with related work.
There is enormous interest in studying how students learn to code, with
and without LLMs:

- BlueJ Blackbox (ICER 2018) has very detailed logs of edit actions. The
  ICER papper lists 18 papers that use the dataset.
- FalconCode: https://huggingface.co/datasets/koutch/falcon_code
- StudentEval (this is LLM related): https://huggingface.co/datasets/wellesley-easel/StudentEval

The datasets above are either open or relatively easy to get access to.

**Questions:**

See weaknesses.

---

> ### Author Response · Authors · 2025-11-18
> **Overall Comment**
>
> Thank you for the review and helpful feedback! We are encouraged that the reviewer found the dataset to be potentially very interesting. We plan on releasing the dataset through a gated repository to protect student privacy while facilitating future research. We address the two weaknesses in separate comments below. Please let us know if there are any further questions we can answer.

---

> ### Author Response · Authors · 2025-11-18
> **Prompt Baselines**
>
> We thank the reviewer for the suggestion to consider prompt-based baselines. The reason we did not include them in our paper was because our primary goal was to understand what can be learned when training on student trace data (versus training on last states only), which can be asked of any base model. Whether a prompting-based approach outperforms our model (unlikely, see below) or not does not affect our core research question.
>
> To illustrate that prompt-based approaches do not already have strong trace modeling abilities, we additionally experimented with **prompting an existing strong pretrained model, StarCoderBase**, which has ~15B parameters and was trained on 1 trillion tokens from GitHub. We prompted this model with few-shot examples to facilitate in-context learning and found that the model ends up copying from context rather than producing meaningful generalizations in three settings we tested:
>
> 1. **Cannot generalize to new program names**: When given a program name not included as a few-shot example, the model cannot generalize to write code for the new program name. For example, in both **examples A and B** below, the model outputs variants of the few shot example for program name colors for the new program names *house* and *star*.
>
> 2. **Cannot make meaningful trace edits**: We also experimented with fixing a program name (i.e. to star) and giving StarCoderBase few-shot examples of different traces for that program name, then prompting it to complete the coding trace given a real student’s first program in another *star* trace. Again, we find in both **examples C and D** below that the model only changes timestamps, repeating code from the first program in the trace (rather than making edits to the code).
>
> 3. **Cannot recover from errors**: In response to the reviewer’s explicit suggestion about error recovery, we also experiment with giving the pretrained StarCoder model a trace with a program that has an error to see whether the model can use ICL to recover from the error. (We few-shot prompt with examples of error recovery). As shown in **example E**, we find that the model cannot fix the bug in the given trace and instead again repeats the given code. As shown in **example F**, even with GPT-5, a powerful closed-weights model, we find that the model cannot fix the bug in the trace, and even generates an unrelated program within the trace.
>
> As shown by these qualitative examples, the fact that we cannot even elicit meaningful generations/completions from the model, despite ICL working for other kinds of programming tasks, highlights that prompt-based models are not well-suited for modeling student code. This aligns with concurrent work by [Miroyan et al., 2025](https://arxiv.org/abs/2507.12674) showing that finetuning models on student code trajectories significantly improves alignment with real student code trajectories over prompting.
>
> That said, we thank the reviewer for raising this question about prompt-based models, and we will be sure to include the above examples in the Appendix of the paper.

---

> > ### Comment · Reviewer_s2Vq · 2025-11-27
> >
> > - I don't think experiments on StarCoderBase are meaningful any more. I would encourage using a contemporary, instruction-following model. E.g., Qwen 2.5 Coder 14B would be more compelling.
> >
> > - The examples below are really hard to follow. The prompt for Example F has format errors. I also cannot tell what the expected output should be.

---

> ### Author Response · Authors · 2025-11-18
> **Related Work**
>
> Thank you for the pointers to the related work!
>
> **BlueJ Blackbox** is indeed a closely related dataset, but there are a few reasons that make it challenging to model similar to our approach. First, PencilCode has a significant set of examples of many students solving the same assignment, identifiable by its name. While Blackbox does have a small set of common assignments, there seems to be little follow-up analysis work on how different students approach the same assignment, which is important for diverse student modeling.
>
> Furthermore, Blackbox data captures all IDE-level actions, which are extremely granular and would likely result in much longer student traces than those in PencilCode. These edits also do not allow for a clean notion of when groups of edits should be considered a separate “program” in a trace. In contrast, the traces in our Pencil Code dataset use program execution to determine each state of the trace, so groups of edits correspond to meaningful attempts made by students before executing.
>
> The **FalconCode** dataset only consists of students' final submissions (not intermediate compilation states), and the StudentEval dataset consists of student’s text prompts for 8 assignments, versus capturing actual coding behavior. While these datasets are not appropriate for  the experiments in our work, we completely agree with the reviewer on their relevance and have updated the introduction and related works sections to reflect this.

---

> > ### Author Response · Authors · 2025-11-18
> > **Example A**
> >
> > **PROMPT**:
> >
> > ```
> > You are an AI model that is optimized to predict what students learning to code on a platform called PencilCode will write.
> >
> > program name: spiral
> >
> > ```CODE 1 (2018-12-10 15:36:36):
> > ht()
> > forever ->
> >   pen random color
> >   rt 10, 5
> >   scale 1.02```
> >
> > -----
> >
> > program name: catalog
> >
> > ```CODE 1 (2017-11-15 15:14:02):
> > speed 100
> > rt 90
> > for color in [red, gold, green, blue]
> >   jump 40, -160
> >   for sides in [3..6]
> >     pen path
> >     for [1..sides]
> >       fd 100 / sides
> >       lt 360 / sides
> >     fill color
> >     fd 40```
> >
> > -----
> >
> > program name: frame
> >
> > ```CODE 1 (2023-09-22 02:50:34):
> > jumpto 50, 100
> > rt 30
> > fd 30
> > pen red, 10
> > lt 360,30
> > jumpto -100, -100
> > pen green, 10
> > lt 60
> > lt 240,60
> > jumpto 50, 50
> > fd 80
> > pen purple, 10
> > lt 330,30
> > lt 90,30
> > fd 60
> > bk 90```
> >
> > -----
> >
> > program name: caterpillarannie
> >
> > ```CODE 1 (2022-02-11 19:08:54):
> > speed 5
> > wear purple
> > jumpto 560,-490
> > dot peachpuff, 100
> > jumpto 520,-470
> > dot pink, 100
> > jumpto 480,-450
> > dot lightskyblue, 100
> > jumpto 440,-430
> > dot peachpuff, 100
> > jumpto 400,-410
> > dot pink, 100
> > jumpto 360,-390
> > dot lightskyblue, 100
> > jumpto 320,-370
> > dot peachpuff, 100
> > jumpto 290,-360
> > dot plum, 100
> > jumpto 300,-350
> > dot indigo, 15
> > jumpto 260,-350
> > dot indigo, 15
> > pen green, 10
> > jumpto 300,-307
> > rt 60,40
> > dot royalblue, 20
> > jumpto 260, -315
> > turnto 0
> > pen green, 10
> > lt 60,40
> > dot royalblue, 20
> > pen red, 10
> > jumpto 260,-375
> > turnto 180
> > lt 180,20```
> >
> > -----
> >
> > program name: colors
> >
> > ```CODE 1 (2017-01-04 18:55:11):
> > dot midnightblue, 900
> > dot indigo, 800
> > dot navy, 700
> > dot olive, 600
> > dot gold, 500
> > dot darkorange, 400
> > dot crimson, 300
> > dot rosybrown, 200
> > dot mistyrose, 100
> > dot black, 50```
> >
> > -----```
> >
> > program name: house
> > ```
> >
> > **GENERATION**:
> >
> > ```
> > ```CODE 1 (2017-01-04 18:55:11):
> > speed 100
> > pen black, 10
> > jumpto 0, 0
> > dot blue, 100
> > dot red, 100
> > dot green, 100
> > dot yellow, 100
> > dot blue, 100
> > dot red, 100
> > dot green, 100
> > dot yellow, 100
> > dot blue, 100
> > dot red, 100
> > dot green, 100
> > dot yellow, 100
> > dot blue, 100
> > dot red, 100
> > dot green, 100
> > dot yellow, 100
> > dot blue, 100
> > dot red, 100
> > dot green, 100
> > dot yellow, 100
> > dot blue, 100
> > dot red, 100
> > dot green, 100
> > dot yellow, 100
> > dot blue, 100
> > dot red, 100
> > dot green, 100
> > dot yellow, 100
> > dot blue, 100
> > dot red, 100
> > dot green, 100
> > dot yellow,
> > ```

---

> > > ### Author Response · Authors · 2025-11-18
> > > **Example B**
> > >
> > > **PROMPT**:
> > >
> > > ```
> > > You are an AI model that is optimized to predict what students learning to code on a platform called PencilCode will write.
> > > -----
> > >
> > > program name: spiral
> > >
> > > ```CODE 1 (2018-12-10 15:36:36):
> > > ht()
> > > forever ->
> > >   pen random color
> > >   rt 10, 5
> > >   scale 1.02```
> > >
> > > -----
> > >
> > > program name: catalog
> > >
> > > ```CODE 1 (2017-11-15 15:14:02):
> > > speed 100
> > > rt 90
> > > for color in [red, gold, green, blue]
> > >   jump 40, -160
> > >   for sides in [3..6]
> > >     pen path
> > >     for [1..sides]
> > >       fd 100 / sides
> > >       lt 360 / sides
> > >     fill color
> > >     fd 40```
> > >
> > > -----
> > >
> > > program name: frame
> > >
> > > ```CODE 1 (2023-09-22 02:50:34):
> > > jumpto 50, 100
> > > rt 30
> > > fd 30
> > > pen red, 10
> > > lt 360,30
> > > jumpto -100, -100
> > > pen green, 10
> > > lt 60
> > > lt 240,60
> > > jumpto 50, 50
> > > fd 80
> > > pen purple, 10
> > > lt 330,30
> > > lt 90,30
> > > fd 60
> > > bk 90```
> > >
> > > -----
> > >
> > > program name: caterpillarannie
> > >
> > > ```CODE 1 (2022-02-11 19:08:54):
> > > speed 5
> > > wear purple
> > > jumpto 560,-490
> > > dot peachpuff, 100
> > > jumpto 520,-470
> > > dot pink, 100
> > > jumpto 480,-450
> > > dot lightskyblue, 100
> > > jumpto 440,-430
> > > dot peachpuff, 100
> > > jumpto 400,-410
> > > dot pink, 100
> > > jumpto 360,-390
> > > dot lightskyblue, 100
> > > jumpto 320,-370
> > > dot peachpuff, 100
> > > jumpto 290,-360
> > > dot plum, 100
> > > jumpto 300,-350
> > > dot indigo, 15
> > > jumpto 260,-350
> > > dot indigo, 15
> > > pen green, 10
> > > jumpto 300,-307
> > > rt 60,40
> > > dot royalblue, 20
> > > jumpto 260, -315
> > > turnto 0
> > > pen green, 10
> > > lt 60,40
> > > dot royalblue, 20
> > > pen red, 10
> > > jumpto 260,-375
> > > turnto 180
> > > lt 180,20```
> > >
> > > -----
> > >
> > > program name: colors
> > >
> > > ```CODE 1 (2017-01-04 18:55:11):
> > > dot midnightblue, 900
> > > dot indigo, 800
> > > dot navy, 700
> > > dot olive, 600
> > > dot gold, 500
> > > dot darkorange, 400
> > > dot crimson, 300
> > > dot rosybrown, 200
> > > dot mistyrose, 100
> > > dot black, 50```
> > >
> > > -----
> > >
> > > program name: star
> > > ```
> > >
> > >
> > >
> > > **GENERATION**
> > >
> > > ```
> > > ```CODE 1 (2017-01-04 18:55:11):
> > > dot midnightblue, 900
> > > dot indigo, 800
> > > dot navy, 700
> > > dot olive, 600
> > > dot gold, 500
> > > dot darkorange, 400
> > > dot crimson, 300
> > > dot rosybrown, 200
> > > dot mistyrose, 100
> > > dot black, 50```
> > >
> > > -----
> > >
> > > program name: spiral
> > >
> > > ```CODE 1 (2018-12-10 15:36:36):
> > > ht()
> > > forever ->
> > >   pen random color
> > >   rt 10, 5
> > >   scale 1.02```
> > >
> > > -----
> > >
> > > program name: catalog
> > >
> > > ```CODE 1 (2017-11-15 15:14:02):
> > > speed 100
> > > rt 90
> > > for color in [red, gold, green, blue]
> > >   jump 40, -160
> > >   for sides in [3..6]
> > >     pen path
> > >     for [1..sides]
> > >       fd 100 / sides
> > >       lt 360 / sides
> > >     fill color
> > >     fd 40```
> > > ```

---

> > > > ### Author Response · Authors · 2025-11-18
> > > > **Example C (Part 1)**
> > > >
> > > > **PROMPT**
> > > >
> > > > ```
> > > > You are an AI model that is optimized to predict what students learning to code on a platform called PencilCode will write. You will be shown the starts of several coding traces written for programs named star. Your task is to complete the traces.
> > > > -----
> > > >
> > > > ```CODE 1 (2021-11-02 07:09:58):
> > > > pen blue, 7
> > > > speed 10
> > > >
> > > > for [1...7]
> > > >   fd 70
> > > >   lt 180
> > > >   fd 70
> > > >   lt 180
> > > >   rt 30
> > > > CODE 2 (2021-11-02 07:10:32):
> > > > pen blue, 7
> > > > speed 10
> > > >
> > > > for [1...13]
> > > >   fd 70
> > > >   lt 180
> > > >   fd 70
> > > >   lt 180
> > > >   rt 30```
> > > >
> > > > -----
> > > >
> > > > ```CODE 1 (2022-04-12 17:45:17):
> > > > speed Infinity
> > > > fd 100
> > > > CODE 2 (2022-04-12 17:45:40):
> > > > speed Infinity
> > > > pen black, 10
> > > > pd()
> > > > moveto 100, 50
> > > > CODE 3 (2022-04-12 17:46:25):
> > > > speed Infinity
> > > > pen black, 10
> > > > pd()
> > > > moveto 100, 50
> > > > moveto -100, 50
> > > > CODE 4 (2022-04-12 17:47:13):
> > > > speed Infinity
> > > > pen black, 10
> > > > pd()
> > > > moveto 100, 50
> > > > moveto -100, 50
> > > > movexy -100, -50
> > > > CODE 5 (2022-04-12 17:47:18):
> > > > speed Infinity
> > > > pen black, 10
> > > > pd()
> > > > moveto 100, 50
> > > > moveto -100, 50
> > > > movexy -100, 50
> > > > CODE 6 (2022-04-12 17:47:20):
> > > > speed Infinity
> > > > pen black, 10
> > > > pd()
> > > > moveto 100, 50
> > > > moveto -100, 50
> > > > movexy -100, 100
> > > > CODE 7 (2022-04-12 17:47:25):
> > > > speed Infinity
> > > > pen black, 10
> > > > pd()
> > > > moveto 100, 50
> > > > moveto -100, 50
> > > > movexy -100, -100
> > > > CODE 8 (2022-04-12 17:47:27):
> > > > speed Infinity
> > > > pen black, 10
> > > > pd()
> > > > moveto 100, 50
> > > > moveto -100, 50
> > > > movexy 100, -100
> > > > CODE 9 (2022-04-12 17:47:40):
> > > > speed Infinity
> > > > pen black, 10
> > > > pd()
> > > > moveto 100, 50
> > > > moveto -100, 50
> > > > movexy 100, -100
> > > > fd 100
> > > > CODE 10 (2022-04-12 17:47:48):
> > > > speed Infinity
> > > > pen black, 10
> > > > pd()
> > > > moveto 100, 50
> > > > moveto -100, 50
> > > > movexy 100, -100
> > > > fd 200
> > > > CODE 11 (2022-04-12 17:48:19):
> > > > speed Infinity
> > > > pen black, 10
> > > > pd()
> > > > moveto 100, 50
> > > > moveto -100, 50
> > > > movexy 100, -100
> > > > fd 200
> > > > moveto 100, 50
> > > > CODE 12 (2022-04-12 17:48:24):
> > > > speed Infinity
> > > > pen black, 10
> > > > pd()
> > > > moveto 100, 50
> > > > moveto -100, 50
> > > > movexy 100, -100
> > > > fd 200
> > > > moveto 100, -50
> > > > CODE 13 (2022-04-12 17:48:32):
> > > > speed Infinity
> > > > pen black, 10
> > > > pd()
> > > > moveto 100, 50
> > > > moveto -100, 50
> > > > movexy 100, -100
> > > > fd 200
> > > > moveto 100, -100
> > > > CODE 14 (2022-04-12 17:48:38):
> > > > speed Infinity
> > > > pen black, 10
> > > > pd()
> > > > moveto 100, 50
> > > > moveto -100, 50
> > > > movexy 100, -100
> > > > fd 200
> > > > moveto -100, 100
> > > > CODE 15 (2022-04-12 17:48:42):
> > > > speed Infinity
> > > > pen black, 10
> > > > pd()
> > > > moveto 100, 50
> > > > moveto -100, 50
> > > > movexy 100, -100
> > > > fd 200
> > > > moveto -100, -100
> > > > CODE 16 (2022-04-12 17:48:45):
> > > > speed Infinity
> > > > pen black, 10
> > > > pd()
> > > > moveto 100, 50
> > > > moveto -100, 50
> > > > movexy 100, -100
> > > > fd 200
> > > > moveto -100, -50
> > > > CODE 17 (2022-04-12 17:48:56):
> > > > speed Infinity
> > > > pen black, 10
> > > > pd()
> > > > moveto 100, 50
> > > > moveto -100, 50
> > > > movexy 100, -100
> > > > fd 200
> > > > moveto -100, -50
> > > > moveto 100, 50
> > > > CODE 18 (2022-04-12 17:49:23):
> > > > speed Infinity
> > > > pen black, 10
> > > > pd()
> > > > moveto 100, 50
> > > > moveto -100, 50
> > > > movexy 100, -100
> > > > fd 200
> > > > moveto -100, -50
> > > > moveto 100, 50
> > > > pu()
> > > > hide()
> > > > CODE 19 (2022-04-12 17:49:48):
> > > > speed 50
> > > > pen black, 10
> > > > pd()
> > > > moveto 100, 50
> > > > moveto -100, 50
> > > > movexy 100, -100
> > > > fd 200
> > > > moveto -100, -50
> > > > moveto 100, 50
> > > > pu()
> > > > hide()
> > > > CODE 20 (2022-04-12 17:49:53):
> > > > speed 40
> > > > pen black, 10
> > > > pd()
> > > > moveto 100, 50
> > > > moveto -100, 50
> > > > movexy 100, -100
> > > > fd 200
> > > > moveto -100, -50
> > > > moveto 100, 50
> > > > pu()
> > > > hide()
> > > > CODE 21 (2022-04-12 17:49:58):
> > > > speed 20
> > > > pen black, 10
> > > > pd()
> > > > moveto 100, 50
> > > > moveto -100, 50
> > > > movexy 100, -100
> > > > fd 200
> > > > moveto -100, -50
> > > > moveto 100, 50
> > > > pu()
> > > > hide()
> > > > CODE 22 (2022-04-12 17:50:02):
> > > > speed 15
> > > > pen black, 10
> > > > pd()
> > > > moveto 100, 50
> > > > moveto -100, 50
> > > > movexy 100, -100
> > > > fd 200
> > > > moveto -100, -50
> > > > moveto 100, 50
> > > > pu()
> > > > hide()
> > > > CODE 23 (2022-04-12 17:53:35):
> > > > speed 15
> > > > pen black, 10
> > > > pd()
> > > > moveto 100, 50
> > > > moveto -100, 50
> > > > movexy 100, -100
> > > > fd 200
> > > > moveto -100, -50
> > > > moveto 100, 50
> > > > fill black
> > > > pu()
> > > > hide()```
> > > >
> > > > -----
> > > >
> > > > ```CODE 1 (2021-12-02 14:24:27):
> > > > pen gold
> > > > fd 100; rt 144
> > > > fd 100; rt 144
> > > > fd 100; rt 144
> > > > fd 100; rt 144
> > > > fd 100; rt 144```
> > > >
> > > > -----
> > > >
> > > > ```CODE 1 (2017-10-26 22:52:46):
> > > > (new Audio('https://upload.wikimedia.org/wikipedia/commons/1/11/06_-_Vivaldi_Summer_mvt_3_Presto_-_John_Harrison_violin.ogg')).play()
> > > > for [1..45]
> > > >   box pink, 700
> > > >   box green, 400
> > > >   for [1..30]
> > > >     speed 200
> > > >     pen purple, 10
> > > >     rt 100, 30
> > > >     rt 135
> > > >   dot grey, 300
> > > >   dot green, 200
> > > >   dot blue, 100
> > > > box pink, 700
> > > > jumpto -270, -148
> > > > rt 30
> > > > pen purple, 10
> > > > fd 400
> > > > rt 120
> > > > fd 400
> > > > rt 145
> > > > fd 500
> > > > rt 145
> > > > fd 500
> > > > rt 155
> > > > fd 530
> > > > fill blue
> > > > CODE 2 (2017-10-26 22:54:20):
> > > > (new Audio('https://upload.wikimedia.org/wikipedia/commons/1/11/06_-_Vivaldi_Summer_mvt_3_Presto_-_John_Harrison_violin.ogg')).play()
> > > > for [1..45]
> > > >   box pink, 700
> > > >   box green, 400
> > > >   for [1..30]
> > > >     speed 200
> > > >     pen purple, 10
> > > >     rt 100, 30
> > > >     rt 135
> > > >   dot grey, 300
> > > >   dot green, 200
> > > >   dot blue, 100
> > > > box pink, 700
> > > > jumpto -170, 201
> > > > rt 30
> > > > pen purple, 10
> > > > fd 400
> > > > rt 120
> > > > fd 400
> > > > rt 145
> > > > fd 500
> > > > rt 145
> > > > fd 500
> > > > rt 155
> > > > fd 530
> > > > fill blue
> > > > CODE 3 (2017-10-26 22:56:54):
> > > > (new Audio('https://upload.wikimedia.org/wikipedia/commons/1/11/06_-_Vivaldi_Summer_mvt_3_Presto_-_John_Harrison_violin.ogg')).play()
> > > > for [1..45]
> > > >   box pink, 700
> > > >   box green, 400
> > > >   for [1..30]
> > > >     speed 200
> > > >     pen purple, 10
> > > >     rt 100, 30
> > > >     rt 135
> > > >   dot grey, 300
> > > >   dot green, 200
> > > >   dot blue, 100
> > > > box pink, 700
> > > > jumpto -170, 201
> > > > rt 30
> > > > pen purple, 10
> > > > fd 400
> > > > rt 120
> > > > fd 400
> > > > rt 145
> > > > fd 500
> > > > rt 145
> > > > fd 500
> > > > rt 155
> > > > fd 530
> > > > fill blue
> > > > fill red

---

> > > > > ### Author Response · Authors · 2025-11-18
> > > > > **Example C (Part 2)**
> > > > >
> > > > > **PROMPT CONTINUED**
> > > > > ```
> > > > > CODE 4 (2017-10-26 22:58:43):
> > > > > (new Audio('https://upload.wikimedia.org/wikipedia/commons/1/11/06_-_Vivaldi_Summer_mvt_3_Presto_-_John_Harrison_violin.ogg')).play()
> > > > > for [1..45]
> > > > >   box pink, 700
> > > > >   box green, 400
> > > > >   for [1..30]
> > > > >     speed 200
> > > > >     pen purple, 10
> > > > >     rt 100, 30
> > > > >     rt 135
> > > > >   dot grey, 300
> > > > >   dot green, 200
> > > > >   dot blue, 100
> > > > > box pink, 700
> > > > > jumpto -170, 201
> > > > > rt 30
> > > > > speed 10
> > > > > pen purple, 10
> > > > > fd 400
> > > > > rt 120
> > > > > fd 400
> > > > > rt 145
> > > > > fd 500
> > > > > rt 145
> > > > > fd 500
> > > > > rt 155
> > > > > fd 530
> > > > > fill blue
> > > > > fill red
> > > > > CODE 5 (2017-10-26 23:00:03):
> > > > > (new Audio('https://upload.wikimedia.org/wikipedia/commons/1/11/06_-_Vivaldi_Summer_mvt_3_Presto_-_John_Harrison_violin.ogg')).play()
> > > > > for [1..45]
> > > > >   box pink, 700
> > > > >   box green, 400
> > > > >   for [1..30]
> > > > >     speed 200
> > > > >     pen purple, 10
> > > > >     rt 100, 30
> > > > >     rt 135
> > > > >   dot grey, 300
> > > > >   dot green, 200
> > > > >   dot blue, 100
> > > > > box pink, 700
> > > > > jumpto -170, 201
> > > > > rt 30
> > > > > speed 1
> > > > > pen purple, 10
> > > > > fd 400
> > > > > rt 120
> > > > > fd 400
> > > > > rt 145
> > > > > fd 500
> > > > > rt 145
> > > > > fd 500
> > > > > rt 155
> > > > > fd 530
> > > > > fill blue
> > > > > fill red
> > > > > CODE 6 (2017-10-26 23:05:32):
> > > > > (new Audio('https://upload.wikimedia.org/wikipedia/commons/1/11/06_-_Vivaldi_Summer_mvt_3_Presto_-_John_Harrison_violin.ogg')).play()
> > > > >
> > > > > jumpto -170, 201
> > > > > rt 30
> > > > > pen purple, 10
> > > > > fd 400
> > > > > rt 120
> > > > > fd 400
> > > > > rt 145
> > > > > fd 500
> > > > > rt 145
> > > > > fd 500
> > > > > rt 155
> > > > > fd 530
> > > > > fill blue
> > > > > fill red
> > > > > for [1..50]
> > > > >   box pink, 700
> > > > >   box green, 400
> > > > >   for [1..30]
> > > > >     speed 200
> > > > >     pen purple, 10
> > > > >     rt 100, 30
> > > > >     rt 135
> > > > >   dot grey, 300
> > > > >   dot green, 200
> > > > >   dot blue, 100
> > > > > box pink, 700
> > > > > jumpto -170, 201
> > > > > rt 30
> > > > > pen purple, 10
> > > > > fd 400
> > > > > rt 120
> > > > > fd 400
> > > > > rt 145
> > > > > fd 500
> > > > > rt 145
> > > > > fd 500
> > > > > rt 155
> > > > > fd 530
> > > > > fill blue
> > > > > CODE 7 (2017-10-26 23:08:46):
> > > > > (new Audio('https://upload.wikimedia.org/wikipedia/commons/1/11/06_-_Vivaldi_Summer_mvt_3_Presto_-_John_Harrison_violin.ogg')).play()
> > > > > for [1..50]
> > > > >   box pink, 700
> > > > >   box green, 400
> > > > >   for [1..30]
> > > > >     speed 200
> > > > >     pen purple, 10
> > > > >     rt 100, 30
> > > > >     rt 135
> > > > >   dot grey, 300
> > > > >   dot green, 200
> > > > >   dot blue, 100
> > > > > box pink, 700
> > > > > jumpto -170, 201
> > > > > rt 30
> > > > > pen purple, 10
> > > > > fd 400
> > > > > rt 120
> > > > > fd 400
> > > > > rt 145
> > > > > fd 500
> > > > > rt 145
> > > > > fd 500
> > > > > rt 155
> > > > > fd 530
> > > > > fill blue
> > > > > CODE 8 (2017-10-26 23:11:45):
> > > > > (new Audio('https://upload.wikimedia.org/wikipedia/commons/1/11/06_-_Vivaldi_Summer_mvt_3_Presto_-_John_Harrison_violin.ogg')).play()
> > > > > for [1..50]
> > > > >   box pink, 700
> > > > >   box green, 400
> > > > >   for [1..30]
> > > > >     speed 200
> > > > >     pen purple, 10
> > > > >     rt 100, 30
> > > > >     rt 135
> > > > >   dot grey, 300
> > > > >   dot green, 200
> > > > >   dot blue, 100
> > > > > box pink, 700
> > > > > for [1..30]
> > > > >   speed 200
> > > > >   jumpto -170, 201
> > > > >   rt 30
> > > > >   pen purple, 10
> > > > >   fd 400
> > > > >   rt 120
> > > > >   fd 400
> > > > >   rt 145
> > > > >   fd 500
> > > > >   rt 145
> > > > >   fd 500
> > > > >   rt 155
> > > > >   fd 530
> > > > >   fill blue
> > > > > CODE 9 (2017-10-26 23:15:40):
> > > > > (new Audio('https://upload.wikimedia.org/wikipedia/commons/1/11/06_-_Vivaldi_Summer_mvt_3_Presto_-_John_Harrison_violin.ogg')).play()
> > > > > for [1..45]
> > > > >   box pink, 700
> > > > >   box green, 400
> > > > >   for [1..45]
> > > > >     speed 200
> > > > >     pen purple, 10
> > > > >     rt 100, 30
> > > > >     rt 135
> > > > >   dot grey, 300
> > > > >   dot green, 200
> > > > >   dot blue, 100
> > > > > box pink, 700
> > > > > jumpto -270, -148
> > > > > rt 60
> > > > > pen purple, 10
> > > > > fd 500```
> > > > >
> > > > > -----
> > > > >
> > > > > ```CODE 1 (2019-04-25 17:49:45):
> > > > > pen gold
> > > > > rt 60
> > > > > for [1..5]
> > > > >   fd 100
> > > > >   rt 2*360/5
> > > > > CODE 2 (2019-04-25 17:49:59):
> > > > > pen gold
> > > > > rt 50
> > > > > for [1..5]
> > > > >   fd 100
> > > > >   rt 2*360/5
> > > > > CODE 3 (2019-04-25 17:50:10):
> > > > > pen gold
> > > > > speed 10
> > > > > rt 50
> > > > > for [1..5]
> > > > >   fd 100
> > > > >   rt 2*360/5
> > > > > CODE 4 (2019-04-25 17:50:18):
> > > > > pen gold
> > > > > speed 10
> > > > > rt 55
> > > > > for [1..5]
> > > > >   fd 100
> > > > >   rt 2*360/5
> > > > > CODE 5 (2019-04-25 17:50:30):
> > > > > pen red
> > > > > speed 10
> > > > > rt 55
> > > > > for [1..5]
> > > > >   fd 100
> > > > >   rt 2*360/5```
> > > > >
> > > > > -----
> > > > >
> > > > > ```CODE 1 (2018-01-03 03:04:30):
> > > > > pen gold
> > > > > fd 100; rt 144
> > > > > fd 100; rt 144
> > > > > fd 100; rt 144
> > > > > fd 100; rt 144
> > > > > fd 100; rt 144
> > > > > fd 100
> > > > > ```
> > > > >
> > > > > **GENERATION**
> > > > >
> > > > > ```
> > > > > CODE 2 (2018-01-03 03:04:40):
> > > > > pen gold
> > > > > fd 100; rt 144
> > > > > fd 100; rt 144
> > > > > fd 100; rt 144
> > > > > fd 100; rt 144
> > > > > fd 100; rt 144
> > > > > fd 100
> > > > > CODE 3 (2018-01-03 03:04:48):
> > > > > pen gold
> > > > > fd 100; rt 144
> > > > > fd 100; rt 144
> > > > > fd 100; rt 144
> > > > > fd 100; rt 144
> > > > > fd 100; rt 144
> > > > > fd 100
> > > > > CODE 4 (2018-01-03 03:04:56):
> > > > > pen gold
> > > > > fd 100; rt 144
> > > > > fd 100; rt 144
> > > > > fd 100; rt 144
> > > > > fd 100; rt 144
> > > > > fd 100; rt 144
> > > > > fd 100
> > > > > CODE 5 (2018-01-03 03:05:04):
> > > > > pen gold
> > > > > fd 100; rt 144
> > > > > fd 100; rt 144
> > > > > fd 100; rt 144
> > > > > fd 100; rt 144
> > > > > fd 100; rt 144
> > > > > fd 100
> > > > > CODE 6 (2018-01-03 03:05:12):
> > > > > ```
> > > > >
> > > > > **ACTUAL TRACE COMPLETION**
> > > > > (Student changed 100 -> 200 in last line)
> > > > >
> > > > > ```
> > > > > CODE 2 (2018-01-03 03:04:54):
> > > > > pen gold
> > > > > fd 100; rt 144
> > > > > fd 100; rt 144
> > > > > fd 100; rt 144
> > > > > fd 100; rt 144
> > > > > fd 100; rt 144
> > > > > fd 200
> > > > > ```

---

> > > > > > ### Author Response · Authors · 2025-11-18
> > > > > > **Example D**
> > > > > >
> > > > > > **PROMPT**
> > > > > >
> > > > > > ```
> > > > > > [same as start of Example C]
> > > > > >
> > > > > > ```CODE 1 (2021-04-08 22:53:43):
> > > > > > pen purple, 10
> > > > > > speed 9999999999999999999999999
> > > > > > for [1..5]
> > > > > >   fd 80
> > > > > >   rt 144
> > > > > > ```
> > > > > >
> > > > > > **GENERATION**
> > > > > >
> > > > > > ```
> > > > > > CODE 2 (2021-04-08 22:54:00):
> > > > > > pen purple, 10
> > > > > > speed 9999999999999999999999999
> > > > > > for [1..5]
> > > > > >   fd 80
> > > > > >   rt 144
> > > > > > CODE 3 (2021-04-08 22:54:10):
> > > > > > pen purple, 10
> > > > > > speed 9999999999999999999999999
> > > > > > for [1..5]
> > > > > >   fd 80
> > > > > >   rt 144
> > > > > > CODE 4 (2021-04-08 22:54:18):
> > > > > > pen purple, 10
> > > > > > speed 9999999999999999999999999
> > > > > > for [1..5]
> > > > > >   fd 80
> > > > > >   rt 144
> > > > > > CODE 5 (2021-04-08 22:54:26):
> > > > > > pen purple, 10
> > > > > > speed 9999999999999999999999999
> > > > > > for [1..5]
> > > > > >   fd 80
> > > > > >   rt 144
> > > > > > CODE 6 (2021-04-08 22:54:34):
> > > > > > pen purple, 10
> > > > > > speed 9999999999999999999999999
> > > > > > for [1..5]
> > > > > >   fd 80
> > > > > >   rt 144
> > > > > > CODE 7 (2
> > > > > > ```
> > > > > >
> > > > > > **ACTUAL TRACE COMPLETION**
> > > > > > ```
> > > > > > CODE 2 (2021-04-08 23:12:24):
> > > > > > pen (random color), 10
> > > > > > speed 9999999999999999999999999
> > > > > > for [1..5]
> > > > > >   fd 80
> > > > > >   rt 144
> > > > > > fill random color
> > > > > > CODE 3 (2021-04-15 12:57:53):
> > > > > > pen ``, 10
> > > > > > speed 9999999999999999999999999
> > > > > > for [1..5]
> > > > > >   fd 80
> > > > > >   rt 144
> > > > > > fill powderblue
> > > > > > CODE 4 (2021-04-15 12:58:11):
> > > > > > pen paleturquoise, 10
> > > > > > speed 9999999999999999999999999
> > > > > > for [1..5]
> > > > > >   fd 80
> > > > > >   rt 144
> > > > > > fill powderblue
> > > > > > ```

---

> > > > > > > ### Author Response · Authors · 2025-11-18
> > > > > > > **Example E**
> > > > > > >
> > > > > > > **PROMPT**
> > > > > > >
> > > > > > > ```
> > > > > > > You are an AI model that is optimized to recover from bugs in code that students learning to code on a platform called PencilCode write. You will be shown the starts of several coding traces. Your task is to fix any existing errors and complete the traces.
> > > > > > > -----
> > > > > > >
> > > > > > > PROGRAM NAME: circle
> > > > > > >
> > > > > > > START:
> > > > > > > ```CODE 1 (2022-11-29 15:05:07):
> > > > > > > lt 360, 90
> > > > > > > CODE 2 (2022-11-29 15:05:43):
> > > > > > > #Blake
> > > > > > > #11/29/22
> > > > > > > #circle
> > > > > > > pen blue
> > > > > > > lt 360, 90
> > > > > > > CODE 3 (2022-11-29 15:05:53):
> > > > > > > #Blake
> > > > > > > #11/29/22
> > > > > > > #circle
> > > > > > > pen blue
> > > > > > > lt 360, 90
> > > > > > > fill blue
> > > > > > > CODE 4 (2022-11-29 15:06:58):
> > > > > > > #Blake
> > > > > > > #11/29/22
> > > > > > > #circle
> > > > > > > pen orange
> > > > > > > rt 360, 90
> > > > > > > fill saddlebrown
> > > > > > > CODE 5 (2022-11-29 15:07:08):
> > > > > > > #Blake
> > > > > > > #11/29/22
> > > > > > > #circle
> > > > > > > pen orange, 10
> > > > > > > rt 360, 90
> > > > > > > fill saddlebrown
> > > > > > > CODE 6 (2022-11-29 15:07:55):
> > > > > > > #Blake
> > > > > > > #11/29/22
> > > > > > > #circle
> > > > > > > pen green, 10
> > > > > > > rt 360, 90
> > > > > > > fill mint```
> > > > > > >
> > > > > > > ERROR RECOVERY COMPLETION:
> > > > > > > ```CODE 7 (2022-11-29 15:08:18):
> > > > > > > #Blake
> > > > > > > #11/29/22
> > > > > > > #circle
> > > > > > > pen green, 10
> > > > > > > rt 360, 90
> > > > > > > fill emerald
> > > > > > > CODE 8 (2022-11-29 15:08:41):
> > > > > > > #Blake
> > > > > > > #11/29/22
> > > > > > > #circle
> > > > > > > pen green, 10
> > > > > > > rt 360, 90
> > > > > > > fill shamrock
> > > > > > > CODE 9 (2022-11-29 15:08:50):
> > > > > > > #Blake
> > > > > > > #11/29/22
> > > > > > > #circle
> > > > > > > pen green, 10
> > > > > > > rt 360, 90
> > > > > > > fill green
> > > > > > > CODE 10 (2022-11-29 15:09:32):
> > > > > > > #Blake
> > > > > > > #11/29/22
> > > > > > > #circle
> > > > > > > pen green, 10
> > > > > > > rt 360, 90
> > > > > > > fill green
> > > > > > > rt 90
> > > > > > > fd 50
> > > > > > > pen brown
> > > > > > > rt 360, 60
> > > > > > > fill saddlebrown```
> > > > > > >
> > > > > > > -----
> > > > > > >
> > > > > > > PROGRAM NAME: loop
> > > > > > >
> > > > > > > START:
> > > > > > > ```CODE 1 (2023-03-07 14:57:40):
> > > > > > > keydown 'w'
> > > > > > > for [1..3]
> > > > > > >   write 'many men'
> > > > > > > CODE 2 (2023-03-07 15:04:36):
> > > > > > > keydown 'w'
> > > > > > > for [1...3]
> > > > > > >   write 'many men'
> > > > > > > CODE 3 (2023-03-07 15:04:48):
> > > > > > > keydown 'w'
> > > > > > > for [0..100]
> > > > > > >   write 'many men'
> > > > > > > CODE 4 (2023-03-07 15:05:06):
> > > > > > > keydown 'w'
> > > > > > > for [0..100]
> > > > > > >   write 'many men'
> > > > > > >   size 100
> > > > > > > CODE 5 (2023-03-07 15:05:17):
> > > > > > > keydown 'w'
> > > > > > > for [0..100]
> > > > > > >   write 'many men'
> > > > > > >   scale 100
> > > > > > > CODE 6 (2023-03-07 15:05:53):
> > > > > > > keydown 'w'
> > > > > > > for [0..100]
> > > > > > >   write 'many men'
> > > > > > >   pensize 50```
> > > > > > >
> > > > > > > ERROR RECOVERY COMPLETION:
> > > > > > > ```CODE 7 (2023-03-07 15:06:02):
> > > > > > > keydown 'w'
> > > > > > > for [0..100]
> > > > > > >   write 'many men'
> > > > > > >   pen 50```
> > > > > > >
> > > > > > > -----
> > > > > > >
> > > > > > > program name: random
> > > > > > >
> > > > > > > START:
> > > > > > > ```CODE 1 (2017-02-07 19:18:14):
> > > > > > > speed 875648
> > > > > > > for [1..4000]
> > > > > > >   moveto random position
> > > > > > >   dot random color
> > > > > > > CODE 2 (2017-02-07 19:18:28):
> > > > > > > rt 90
> > > > > > > for [1..10]
> > > > > > >   dot blue
> > > > > > >   for [1..4]
> > > > > > >     fd 50
> > > > > > >     rt 90
> > > > > > >   lt 36
> > > > > > >   bk 50
> > > > > > > CODE 3 (2017-02-07 19:20:01):
> > > > > > > for [1..9000]
> > > > > > >   moveto random position
> > > > > > >   dot random color
> > > > > > > CODE 4 (2017-02-07 19:20:11):
> > > > > > > speed 1000
> > > > > > > for [1..9000]
> > > > > > >   moveto random position
> > > > > > >   dot random color
> > > > > > > CODE 5 (2017-02-07 19:20:45):
> > > > > > > speed 1000
> > > > > > > for [1..9000]
> > > > > > >   moveto random position
> > > > > > >   dot random color
> > > > > > >   speed 1000
> > > > > > > for [1..9000]
> > > > > > >   moveto random position
> > > > > > >   dot random color
> > > > > > >   speed 1000
> > > > > > > for [1..9000]
> > > > > > >   moveto random position
> > > > > > >   dot random color
> > > > > > > CODE 6 (2017-02-07 21:05:21):
> > > > > > > speed 1000000000
> > > > > > > for [1..9000]
> > > > > > >   moveto random position
> > > > > > >   dot random color [skyblue, palegreen, salmon]
> > > > > > >   speed 1000
> > > > > > > for [1..9000]
> > > > > > >   moveto random position
> > > > > > >   dot random color [skyblue, palegreen, salmon]
> > > > > > >   speed 1000
> > > > > > > for [1..9000]
> > > > > > >   moveto random position
> > > > > > >   dot random color [skyblue, palegreen, salmon]```
> > > > > > >
> > > > > > > ERROR RECOVERY COMPLETION:
> > > > > > > ```
> > > > > > >
> > > > > > >
> > > > > > > **GENERATION**
> > > > > > >
> > > > > > > ```
> > > > > > > ```CODE 7 (2017-02-07 21:05:30):
> > > > > > > speed 1000000000
> > > > > > > for [1..9000]
> > > > > > >   moveto random position
> > > > > > >   dot random color [skyblue, palegreen, salmon]
> > > > > > >   speed 1000
> > > > > > > for [1..9000]
> > > > > > >   moveto random position
> > > > > > >   dot random color [skyblue, palegreen, salmon]
> > > > > > >   speed 1000
> > > > > > > for [1..9000]
> > > > > > >   moveto random position
> > > > > > >   dot random color [skyblue, palegreen, salmon]
> > > > > > > CODE 8 (2017-02-07 21:05:40):
> > > > > > > speed 1000000000
> > > > > > > for [1..9000]
> > > > > > >   moveto random position
> > > > > > >   dot random color [skyblue, palegreen, salmon]
> > > > > > >   speed 1000
> > > > > > > for [1..9000]
> > > > > > >   moveto random position
> > > > > > >   dot random color [skyblue, palegreen, salmon]
> > > > > > >   speed 1000
> > > > > > > for [1..9000]
> > > > > > >   moveto random position
> > > > > > >   dot random color [skyblue, palegreen, salmon]
> > > > > > > CODE 9 (2017-02-07 21:05:50):
> > > > > > > speed 1000000000
> > > > > > > for [1..9000]
> > > > > > >   moveto random position
> > > > > > >   dot random color [skyblue, palegreen, salmon]
> > > > > > >   speed 1000
> > > > > > > for [1..9000]
> > > > > > >   moveto random position
> > > > > > >   dot random color [skyblue, palegreen, salmon]
> > > > > > >   speed 1000
> > > > > > > ```
> > > > > > >
> > > > > > > **ACTUAL TRACE COMPLETION (with Error Recovery)**
> > > > > > >
> > > > > > > ```
> > > > > > > CODE 7 (2017-02-07 21:05:32):
> > > > > > > speed 1000000000
> > > > > > > for [1..9000]
> > > > > > >   moveto random position
> > > > > > >   dot random color, [skyblue, palegreen, salmon]
> > > > > > >   speed 1000
> > > > > > > for [1..9000]
> > > > > > >   moveto random position
> > > > > > >   dot random color, [skyblue, palegreen, salmon]
> > > > > > >   speed 1000
> > > > > > > for [1..9000]
> > > > > > >   moveto random position
> > > > > > >   dot random color, [skyblue, palegreen, salmon]
> > > > > > > CODE 8 (2017-02-07 22:39:13):
> > > > > > > speed 100
> > > > > > > for [1..9000]
> > > > > > >   moveto random position
> > > > > > >   dot random color, [skyblue, palegreen, salmon]
> > > > > > >   speed 1000
> > > > > > > for [1..9000]
> > > > > > >   moveto random position
> > > > > > >   dot random color, [skyblue, palegreen, salmon]
> > > > > > >   speed 1000
> > > > > > > for [1..9000]
> > > > > > >   moveto random position
> > > > > > >   dot random color, [skyblue, palegreen, salmon]
> > > > > > > ```

---

> > > > > > > > ### Author Response · Authors · 2025-11-18
> > > > > > > > **Example F**
> > > > > > > >
> > > > > > > > **PROMPT**:
> > > > > > > >
> > > > > > > > ```
> > > > > > > > You are an AI model that is optimized to recover from bugs in code that students learning to code on a platform called PencilCode write. You will be shown the starts of a coding trace. Your task is to fix any existing errors and complete the trace.
> > > > > > > >
> > > > > > > > CODE 1 (2017-02-07 19:18:14): speed 875648 for [1..4000] moveto random position dot random color CODE 2 (2017-02-07 19:18:28): rt 90 for [1..10] dot blue for [1..4] fd 50 rt 90 lt 36 bk 50 CODE 3 (2017-02-07 19:20:01): for [1..9000] moveto random position dot random color CODE 4 (2017-02-07 19:20:11): speed 1000 for [1..9000] moveto random position dot random color CODE 5 (2017-02-07 19:20:45): speed 1000 for [1..9000] moveto random position dot random color speed 1000 for [1..9000] moveto random position dot random color speed 1000 for [1..9000] moveto random position dot random color CODE 6 (2017-02-07 21:05:21): speed 1000000000 for [1..9000] moveto random position dot random color [skyblue, palegreen, salmon] speed 1000 for [1..9000] moveto random position dot random color [skyblue, palegreen, salmon] speed 1000 for [1..9000] moveto random position dot random color [skyblue, palegreen, salmon]
> > > > > > > > ```
> > > > > > > >
> > > > > > > > **GENERATION**
> > > > > > > > ```
> > > > > > > > CODE 7 (2017-02-07 21:06:03):
> > > > > > > > speed 1000000000
> > > > > > > > for [1..12000]
> > > > > > > >   moveto random position
> > > > > > > >   dot random color [skyblue, palegreen, salmon, gold, plum]
> > > > > > > >
> > > > > > > > CODE 8 (2017-02-07 21:06:27):
> > > > > > > > rt 45
> > > > > > > > for [1..20]
> > > > > > > >   dot gold
> > > > > > > >   for [1..8]
> > > > > > > >     fd 40
> > > > > > > >     rt 45
> > > > > > > >   lt 18
> > > > > > > >   bk 40
> > > > > > > >
> > > > > > > > CODE 9 (2017-02-07 21:07:10):
> > > > > > > > speed 500000000
> > > > > > > > for [1..15000]
> > > > > > > >   moveto random position
> > > > > > > >   dot random color [salmon, skyblue, palegreen, gold, orchid]
> > > > > > > >
> > > > > > > > CODE 10 (2017-02-07 21:07:49):
> > > > > > > > speed 1000
> > > > > > > > for [1..3]
> > > > > > > >   for [1..6000]
> > > > > > > >     moveto random position
> > > > > > > >     dot random color [skyblue, salmon, palegreen]
> > > > > > > >   rt 120
> > > > > > > > ```

---

> > > > > > > > > ### Author Response · Authors · 2025-11-18
> > > > > > > > > **Any remaining questions?**
> > > > > > > > >
> > > > > > > > > We hope the additional prompt-based experiments and discussion of related work have addressed the reviewer’s key concerns. We are grateful for your feedback and would be happy to elaborate on any remaining questions.

---

> ### Author Response · Authors · 2025-11-27
> **Prompt Baselines**
>
> 1. To clarify any misunderstandings, Example F was with GPT-5, a more contemporary, instruction-following model. There are a couple of failure modes.
>
> First, if you look at Code 8, you can see the program completely deviate from the other programs in the state, including more navigation commands and without any random color selection. This behavior was not in-context, is not seen in any student data (program traces should track the development of a program for the same assignment), and would be addressed by fine-tuning.
>
> Second, the concrete error it should recover is with the parameters and usage of dot, by putting a comma after color.  ```(dot random color  [skyblue, palegreen, salmon] --> dot random color,  [skyblue, palegreen, salmon])```. We are unable to elicit this fix from any contemporary model, including GPT-5.
>
> Finally, if the reviewer's formatting concerns are simply spacing, that is just improper copying of the prompt to OpenReview. They were correct in the prompt, as demonstrated by the generated output carrying proper indentation.
>
> 2. Our examples still fail even when using powerful code models like Qwen3-Coder. This also aligns with concurrent work by [Miroyan et al., 2025](https://arxiv.org/abs/2507.12674) showing that a pretrained Qwen2.5 Coder 7B model underperforms finetuned models at predicting student code trajectories, even for Python.
>
> With example A, the model states that it is generating a house, but then generates  code where the roof triangle is not attached to the bottom rectangle:
>
> ```speed 5
> jumpto -100, -100
> pen blue, 3
> for [1..4]
>   fd 200
>   rt 90
> jumpto -100, 100
> pen red, 3
> for [1..3]
>   fd 200
>   rt 120
> jumpto -30, -100
> pen brown, 3
> for [1..2]
>   fd 60
>   rt 90
>   fd 80
>   rt 90
> jumpto 50, -50
> dot yellow, 10
> ```
>
>
> Furthermore, the examples are not similar to the actual student assignments. For example B, Qwen3 generates code that creates a 5 pointed star (that itself is incorrectly rendered):
> ```speed 5
> pen blue, 3
> jumpto 0, 100
> for [1..5]
>   fd 200
>   rt 144
> jumpto 0, 100
> pen yellow, 2
> for [1..5]
>   fd 180
>   rt 144
> fill yellow
> ```
>
> But for real students on Pencil Code, the star assignment is intended to be a lesson about functions, and students should be writing code that generates n stars for a given parameter, like:
> ```star = (n) ->
>   for [1..n]
>     fd 100
>     rt 2 * 360 / n
> pen turquoise, 10
> star 7
> ```
>
> In order for prompting to be a suitable baseline for our work, large pre-trained models need to be able to capture the **intent behind assignments students solved** and a **strong understanding of potential errors**. From our results, this is currently best only captured by finetuning on student data.
>
> Nevertheless, as stated in our rebuttal, our primary goal was to understand what can be learned when training on student trace data (versus training on last states only), which is not affected if a strong prompt baseline ever does exist.

---

### Author Response · Authors · 2025-12-02
**Note to the ACs**

Dear ACs,

Thank you for your time supporting the reviewing process! We summarize the biggest concerns raised by reviewers and our response, as well as a list of our revisions. Importantly, we believe there were some misconceptions about our experiments that our rebuttal sought to clarify.

**Core Review Concerns**

1. **Outdated Models/Baselines (s2Vq, ngbA, ncN7)**: Reviewers suggested using SOTA models (Qwen, StarCoder, Llama) as baselines. We clarify that our **research question is model-agnostic** and fundamentally about *training data* (traces vs. synthetic vs. final-only), not achieving SOTA performance. Our controlled experiments with **GPT-2 and OLMo-2** demonstrate that *findings hold across scale*, with gaps widening at larger sizes.


    During rebuttal, we have further provided [extensive evidence](https://openreview.net/forum?id=OndDxNGrqJ&noteId=ZkbGkE3Sbc) showing that even **powerful pretrained models (15B StarCoderBase, GPT-5)** *cannot effectively model student traces through prompting* alone: they *copy context*, *fail to generalize to new assignments*, and *cannot recover from errors*. This supports our approach of finetuning on student data. Concurrent work [(Miroyan et al., 2025)](https://arxiv.org/abs/2507.12674) also confirms that finetuning on student trajectories significantly outperforms prompting even for Qwen models.


2. **Education Implications (fQu1, ngbA)**: Reviewers asked about practical educational applications. We [clarify](https://openreview.net/forum?id=OndDxNGrqJ&noteId=NN0dJYBxt0) that modeling student behavior is a necessary foundation for downstream educational systems. Furthermore, Section 4.5 shows a concrete application in **code error recovery** with *personalized style preservation*. We have **expanded Section 6** to *explicitly discuss integration with intelligent tutoring systems, hint generation, and process-based assessment applications*.

3. **Student Embedding / Generalization (fQu1)**: We believe the concerns surrounding student embeddings and generalization were possible **misconceptions** that we hope we have [clarified via engagement with Review fQu1](https://openreview.net/forum?id=OndDxNGrqJ&noteId=fmIHZIYGwT). Student IDs serve as *indices for learned behavioral embeddings* (like word tokens in LMs), *not features themselves*. Our claim is not zero-shot inference about unseen students, but that: (a) embeddings capture important dimensions of variation, (b) this enables efficient personalization with 2-4 traces (Section 4.4), and (c) these representations predict future behavior within and across traces.

4. **Scope Too Large/Too Narrow (ngbA)**: Reviewer ngbA felt we addressed "too many research questions," while appreciating the dataset contribution. We [clarify](https://openreview.net/forum?id=OndDxNGrqJ&noteId=Lkrj71uVTg) that our work explores **three core contributions deeply**: (1) *trace-trained models learn richer representations* (4.1-4.3), (2) representations enable *efficient personalization* (4.4), and (3) *practical educational applications like error recovery* (4.5). Each section examines multiple controls. While limited to one platform, PencilCode supports diverse activities across multiple languages and reaches 1M+ students, and we believe this is the first large-scale student trace dataset with high student and assignment density.

Finally, we note that some reviews mentioned issues we were unable to identify (e.g. figure styles/citations in ngbA's review), but were unfortunately unable to get more information from the reviewer. We hope the AC can take this into account.

**Paper Modifications:**

1. **Examples** showing in-context learning with a **powerful code model** (~15B pretrained StarCoderBase model) fail to generalize to new program names, make meaningful trace edits, or recover from errors, supporting our choice to work with finetuned models
2. Revised **Section 4** introduction to **better highlight our 3 core contributions**, and the logical progression of our experiments
3. Expanded discussion in **Section 6** to emphasize **education implications**
4. Updated **Related Works** section to include reviewer's suggested papers
5. Clarification in 4.4 on how **student IDs** are mapped to random embeddings
6. Provided **examples** in Appendix showing our dataset includes **complex/non-trivial assignments** on Pencil Code
7. Moved **dataset details** from Appendix to main text to improve clarity

---

### Meta-Review · Area_Chair_dZmp · 2025-12-29

**Summary:**

Paper Summary. This paper first presents a novel dataset of 3.8 million programming traces from users on a free online educational platform. It then uses this dataset to study how programming traces are useful for training language models to modeling students' programming.

Paper Strengths. This is a highly interesting, meaningful, and potentially very impactful project. The novelty is its focus on programmer behaviors by moving beyond training models to generate final code towards modeling student programming traces. The dataset consisting of 3.8 Million Program Traces is also a unique contribution that can enable many future research in this direction.

Reviewer Concerns. Reviewers have multiple concerns. (1) Missing baselines of missing obvious baselines of prompting with programming traces (Reviewer s2Vq) (2) Discussion with related work (Reviewer s2Vq, Reviewer ncN7) (3) Unclear paper contribution in terms of educational goals and Lack of user study (Reviewer fQu1, Reviewer ncN7). (4) Clarification of student embeddings and generalization to new students (Reviewer fQu1) (5) Outdated model choices (Reviewer ngbA, Reviewer ncN7) (6) Dataset presentation and Narrow and simple data source (Reviewer ngbA, Reviewer ncN7) (7) Data Release (Reviewer s2Vq, Reviewer ncN7)

**Reviewer Concerns:**

Addressed Concerns.
The author provided solid rebuttal and arguments to concerns including (2) (4) (5) (6) (7)

Outstanding Concerns.
By considering all reviewers' comments holistically, I believe the major issue is the unclear scope, positioning, and contribution of this paper. This is a result of the combination of both presentation issues and experiment designs. The presentation issue is largely solved by rebuttal. However, to effectively address this, at least one of these two outstanding concerns must be addressed.

(1) Lack of thorough comparisons with related work and baselines. If the paper is positioned to be a technical contribution of training LLMs with coding traces, then it would be necessary to conduct a thorough comparison with baselines and existing models. While the author provided further experiments based on prompting StarCoderBase, it does not convince the reviewer as a split comparison because it is an outdated coding model and the discussion is purely qualitative by presenting several examples. Some comprehensive quantitative comparisons would be helpful.

(2) Lack of deep discussion of user study or educational implications. If the paper aims to contribute to a practical CS education research, it would be necessary to build systems (e.g., early intervention systems, feedback generation models) or conduct a user study to demonstrate the educational benefits for students learning to program. The current author rebuttal, while making sense, is still at the discussion level.

Furthermore, as a personal suggestion, while this paper is in the datasets and benchmarks track, I believe the dataset itself is not thoroughly discussed, with only a short section 2 with two paragraphs. If the author still wants to position this as a dataset paper, I strongly suggest adding more stats, analysis, comparisons of the dataset to show its uniqueness and complexity, and demonstrate diverse ways of using it beyond training on programming traces.

**Reviewer Scores:**

Reviewer s2Vq: 4 -> 4 (the prompting baseline result is qualitative and not convincing)

Reviewer fQu1: 2 -> 4

Reviewer ngbA: 2 -> 4

Reviewer ncN7: 6 -> 6 (user study still missing, dataset presentation still insufficient in updated pdf)

---

### Decision · Program_Chairs · 2026-01-26

Reject